# Towards a Sharp Analysis of Offline Policy Learning for $f$-Divergence-Regularized Contextual Bandits

**Qingyue Zhao**[1*]**, Kaixuan Ji**[1*]**, Heyang Zhao**[1*]**, Tong Zhang**[2]**, Quanquan Gu**[1]
[1]University of California, Los Angeles
[2]University of Illinois Urbana-Champaign
{kaixuanji,zhaoqy24,hyzhao}@cs.ucla.edu
tozhang@illinois.edu,qgu@cs.ucla.edu

## Abstract

Many offline reinforcement learning algorithms are underpinned by $f$-divergence regularization, but their sample complexity *defined with respect to regularized objectives* still lacks tight analyses, especially in terms of concrete data coverage conditions. In this paper, we study the exact concentrability requirements to achieve the $\widetilde{\Theta}(\epsilon^{-1})$ sample complexity for offline $f$-divergence-regularized contextual bandits. For reverse Kullback–Leibler (KL) divergence, arguably the most commonly used one, we achieve an $\widetilde{O}(\epsilon^{-1})$ sample complexity under single-policy concentrability for the first time via a novel pessimism-based analysis, surpassing existing $\widetilde{O}(\epsilon^{-1})$ bound under all-policy concentrability and $\widetilde{O}(\epsilon^{-2})$ bound under single-policy concentrability. We also propose a near-matching lower bound, demonstrating that a multiplicative dependency on single-policy concentrability is necessary to maximally exploit the curvature property of reverse KL. Moreover, for $f$-divergences with strongly convex $f$, to which reverse KL *does not* belong, we show that the sharp sample complexity $\widetilde{\Theta}(\epsilon^{-1})$ is achievable even without pessimistic estimation or single-policy concentrability. We further corroborate our theoretical insights with numerical experiments and extend our analysis to contextual dueling bandits. We believe these results take a significant step towards a comprehensive understanding of objectives with $f$-divergence regularization.

## 1 Introduction

Due to the data-hungry and instable nature of reinforcement learning (RL), divergences that are straightforward to estimate via Monte Carlo or amenable to constrained optimization stand out from numerous candidates (Rényi, 1961; Csiszár, 1967; Müller, 1997; Basseville, 2013) as regularizers; the former family is typically *$f$-divergence* (Rényi, 1961) because any of them is an expectation, for which empirical average is a good proxy (Levine, 2018; Levine et al., 2020); and the latter class subsumes those with nice positive curvatures (e.g., Bregman divergence (Bregman, 1967) induced by strongly convex functions). In particular, *Kullback-Leibler (KL) divergence* is the only one at the intersection of $f$-divergence and Bregman divergence (Jiao et al., 2014, Theorem 5), indicating its theoretical advantage among common choices from both computational and statistical aspects. Also, the *KL-regularized RL objective* is arguably the most popular one in practice:

$$J(\pi) = \mathbb{E}_\pi[r] - \eta^{-1}\mathsf{KL}(\pi\|\pi^{\mathsf{ref}}), \tag{1.1}$$

where $r$ is the reward, $\pi^{\mathsf{ref}}$ is a reference policy, $\mathsf{KL}(\pi\|\pi^{\mathsf{ref}})$ is the reverse KL divergence, and $\eta > 0$ is the inverse temperature. When $\pi^{\mathsf{ref}}$ is uniform, (1.1) reduces to the entropy-regularized objective that encourages diverse actions and enhances robustness (Williams, 1992; Ziebart et al., 2008; Levine & Koltun, 2013; Levine et al., 2016; Haarnoja et al., 2018; Richemond et al., 2024; Liu et al., 2024). KL regularization has also been widely used in the RL fine-tuning of large language models

---

*Equal Contribution

Table 1: Comparison of sample complexity bounds for finding $\epsilon$-optimal policy for offline contextual bandits with KL- and (strongly convex) $f$-divergence regularization. Constants and $\mathrm{polylog}$ factors are omitted here except the metric entropy $\log \mathcal{N}$. "Reverse-KL" stands for KL-regularized contextual bandits and "$f$-divergence w/ s.c., $f$" for the counterpart with an $\alpha$-strongly convex $f$. The two existing upper bounds are adapted from the implicit form in Xiong et al. (2024, Theorem 3.1) and Zhao et al. (2024, Theorem 3.3 and Theorem 4.4), of which the detailed adaptions are deferred to Appendix A. The relationship between $D_{\pi*}^2$ and $C^{\pi^*}$ is detailed in Section 2.1.

| Regularizer | | Xiong et al. (2024) | Zhao et al. (2024) | This work |
|---|---|---|---|---|
| Reverse KL | Upper | $d\epsilon^{-2}$ | $\eta D^2 \epsilon^{-1} \log \mathcal{N}$ | $\eta D_{\pi*}^2 \epsilon^{-1} \log \mathcal{N}$ |
| | Lower | - | $\eta \epsilon^{-1} \log \mathcal{N}$ | $\eta C^{\pi^*} \epsilon^{-1} \log \mathcal{N}$ |
| $f$-divergence w/ s.c. $f$ | Upper | - | - | $\alpha^{-1} \eta \epsilon^{-1} \log \mathcal{N}$ |
| | Lower | - | - | $\alpha^{-1} \eta \epsilon^{-1} \log \mathcal{N}$ |

(Ouyang et al., 2022; Rafailov et al., 2023), where $\pi^{\mathrm{ref}}$ is the base model. Given its widespread use, there has been a surge of interest in understanding the role of KL regularization in RL by both empirical studies (Ahmed et al., 2019; Liu et al., 2019) and theoretical analysis (Geist et al., 2019; Vieillard et al., 2020; Kozuno et al., 2022). There are also lines of research on KL regularization in online learning (Cai et al., 2020; He et al., 2022; Ji et al., 2023) and convex optimization (Neu et al., 2017). However, most of these works still study the unregularized reward maximization objective, against which the sample complexity is at least $\Omega(\epsilon^{-2})$.[1]

Several recent papers (Xiong et al., 2024; Xie et al., 2024; Zhao et al., 2024; Foster et al., 2025; Aminian et al., 2025) switched the focus to analyzing the sub-optimality defined via the regularized objective (1.1), under which an $\Omega(\epsilon^{-1})$ sample complexity is possible (Zhao et al., 2024, Theorem 3.6). However, even restricted to the pure i.i.d. setting, existing analyses in this vein either result in still $\widetilde{O}(\epsilon^{-2})$ bounds (Xiong et al., 2024; Xie et al., 2024) or has stringent (local) all-policy concentrability dependencies in their upper bounds (Zhao et al., 2024; Aminian et al., 2025).[2] Thus, there are by far no tight bounds in terms of both the dependency of $\epsilon^{-1}$ and data coverage conditions for KL-regularized offline decision making. In addition, all analyses above set KL as the right target by default; but reverse KL is the $f$-divergence with $f(x) = x \log x$, which is merely convex. Therefore, it is also unknown whether $f$-divergence regularizers with even nicer (e.g., strongly convex) $f$, whose performance against the reward maximization objective are provably promising (Zhan et al., 2022; Gabbianelli et al., 2024; Huang et al., 2025b), can enjoy a better coverage dependency in their sample complexity when the corresponding regularized objectives serve as the performance metric. Because data coverage (i.e., concentrability) conditions captures the crucial distributional shift issue in offline RL (Levine et al., 2020), the aforementioned perspectives motivate a pivotal open problem:

> *What is the weakest coverage condition required for* offline learning to be near-optimal with respect to $f$-*divergence-regularized objectives?*

We attack this problem by showing near-optimal sample complexity with matching concentrability dependencies for two representative subclasses of $f$-divergence. First, for contextual bandits with KL regularization, we achieve a near-optimal sample complexity guarantee with linear dependence on *single*-policy coverage ratio. Our novel lower bound further indicates that this multiplicative dependency on *single*-policy concentrability is necessary. Surprisingly, for $f$-divergence with $\alpha$-strongly-convex $f$, we prove nearly matching sample complexity bounds of $\widetilde{\Theta}(\alpha^{-1} \eta \epsilon^{-1})$, eliminating the dependence on coverage for the first time. For the ease of comparison, we adapt existing counterparts under our notation to the offline setting and summarize them in Table 1.

## 1.1 Contributions

- For KL regularization, we propose a pessimism-based algorithm achieving the tight sample complexity under *single*-policy concentrability. We also obtain a lower bound that linearly scale with the density-ratio-based single-policy concentrability. Both results strictly improves upon previous

---

[1]See Appendix A.1 for detailed reasons.
[2]See Section 2.1 for details on coverage conditions.

works (Zhao et al., 2024; Foster et al., 2025) in the offline setting, showing that single-policy concentrability is both sufficient and necessary to achieve the $\widetilde{\Theta}(\epsilon^{-1})$ sample complexity.

- Technically speaking, our analysis exploits the strong convexity of KL and pessimism of the reward estimator, to refine a mean-value-type risk upper bound (Lemma 2.14) to its, which in turn leads to a novel moment-based analysis, effectively bypassing the need for uniform control over the discrepancy between any two functions in the function class. To the best of our knowledge, this machinery has not been used in the standard analysis of existing offline RL algorithms and may be of independent interest.

- For $f$-divergence-regularized objectives with strongly convex $f$, we design a truly lightweight algorithm free of pessimism-based gadgets and still obtain the $\widetilde{\Theta}(\epsilon^{-1})$ sample complexity certified by a matching lower bound without coverage conditions.

- We verify the statistical rates above in numerical experiments, and demonstrate the versatility of all algorithmic and constructive proof ideas above by extending them to $f$-divergence-regularized contextual dueling bandits (CDBs), achieving similar $\widetilde{\Theta}(\epsilon^{-1})$ sample complexity bounds. Moreover, all algorithms are applicable for reward function classes with small metric entropy.

## 1.2  KEY RELATED WORK

We review two key lines of theoretical progress that are relevant to our algorithm design and analysis. **Pessimism in offline RL.** The principle of pessimism has been underpinning offline RL for both the tabular (Rashidinejad et al., 2021) and function approximation (Jin et al., 2021) settings under the name of lower confidence bound (LCB). For contextual bandits, it is behind the adaptively optimal sample complexity analysis (Li et al., 2022). Shi et al. (2022) proposed a LCB-based model-free algorithm for tabular RL with near-optimal guarantee. Jin et al. (2021); Xiong et al. (2022); Di et al. (2024) utilized LCB in conjunction with the classic least-square value iteration paradigm to derive $\widetilde{O}(\epsilon^{-2})$ sample complexity results for model-free RL with function approximation. The line of work from Rashidinejad et al. (2021); Xie et al. (2021b) to Li et al. (2024) settled the sample complexity of tabular model-based RL via pessimistic estimators exploiting the variance information. It is also possible to leverage the idea of pessimism to design model-based algorithms under general function approximation that are at least statistically efficient (Xie et al., 2021a; Uehara & Sun, 2021; Wang et al., 2024). The principle of pessimism has also been applied in counterfactual empirical risk minimization (Swaminathan & Joachims, 2015; London & Sandler, 2019) and offline policy learning (Sakhi et al., 2023; 2024), which are orthogonal to our contributions.

However, in terms of risk decomposition, to the best of our knowledge, none of these pessimism-based analyses really goes beyond the performance difference lemma (Foster & Rakhlin, 2023, Lemma 13) or simulation lemma (Foster & Rakhlin, 2023, Lemma 23); both of which are not able to capture the strong concavity of KL-regularized objectives even in the bandit setting. The algorithmic idea of using pessimistic least-square estimators under general function approximation in Jin et al. (2021); Di et al. (2024) is similar to ours, but their sub-optimality gap is bounded by the sum of bonuses, which cannot directly lead to the desired sample complexity of our objective.

**Offline CDBs.** CDBs (Dudík et al., 2015) is the contextual extension of dueling bandits in the classic literature of online learning from pairwise comparisons (Yue et al., 2012; Zoghi et al., 2014). Since the empirical breakthrough of preference-based RL fine-tuning of LLMs (Ouyang et al., 2022), the theory of offline CDBs has received more attention under linear function approximation (Zhu et al., 2023; Xiong et al., 2024) and general function approximation (Zhan et al., 2022; Zhao et al., 2024; Song et al., 2024; Huang et al., 2025b). Preference models without stochastic transitivity (Munos et al., 2023; Ye et al., 2024; Wu et al., 2024; Zhang et al., 2024) are beyond the scope of this work, namely, our preference labels are assumed to follow the Bradley-Terry Model (Bradley & Terry, 1952).

**Notation.**  The sets $\mathcal{S}$ and $\mathcal{A}$ are assumed to be countable throughout the paper. For nonnegative sequences $\{x_n\}$ and $\{y_n\}$, we write $x_n = O(y_n)$ if $\limsup_{n\to\infty} x_n/y_n < \infty$, $y_n = \Omega(x_n)$ if $x_n = O(y_n)$, and $y_n = \Theta(x_n)$ if $x_n = O(y_n)$ and $x_n = \Omega(y_n)$. We further employ $\widetilde{O}(\cdot), \widetilde{\Omega}(\cdot)$, and $\widetilde{\Theta}$ to hide polylog factors. For countable $\mathcal{X}$ and $\mathcal{Y}$, we denote the family of probability kernels from $\mathcal{X}$ to $\mathcal{Y}$ by $\Delta(\mathcal{Y}|\mathcal{X})$. For $g : \mathcal{X} \to \mathbb{R}$, its infinity norm is denoted by $\|g\|_\infty := \sup_{x\in\mathcal{X}} |g(x)|$. For a pair of probability measures $P \ll Q$ on the same space and function $f : \mathbb{R}_+ \to \mathbb{R}$, their

$f$-divergence is $D_f(P\|Q) := \int f(\,\mathrm{d}P/\,\mathrm{d}Q)\,\mathrm{d}Q$. Specifically, when $f(x) = x\log x$, $f$-divergence becomes KL divergence denoted as $\mathsf{KL}(P\|Q) := \int \log(\,\mathrm{d}P/\,\mathrm{d}Q)\,\mathrm{d}P$, and when $f(x) = |x-1|/2$, it becomes the total variation (TV) distance, which is denoted as $\mathsf{TV}(P\|Q) := 0.5\int |\,\mathrm{d}P - \,\mathrm{d}Q|$. We use $\mathrm{supp}(P)$ to denote the support set of $P$.

## 2 KL-REGULARIZED CONTEXTUAL BANDITS

In this section, we introduce a pessimism-based algorithm, PCB-KL, for offline KL-regularized contextual bandits. We then showcase our novel analysis techniques for PCB-KL, which couples the algorithmic pessimism with the curvature property of KL-regularized objectives.

### 2.1 PROBLEM SETUP

We consider contextual bandit, which is denoted by a tuple $(\mathcal{S}, \mathcal{A}, r, \pi^{\mathsf{ref}})$. Specifically, $\mathcal{S}$ is the context space, $\mathcal{A}$ is the action space and $r : \mathcal{S} \times \mathcal{A} \to [0,1]$ is the reward function. In the offline setting, the agent only has access to an i.i.d. dataset $\mathcal{D} = \{(s_i, a_i, r_i)\}_{i=1}^n$. Here $s_i's$ are states sampled from $\rho \in \Delta(\mathcal{S})$, $a_i \in \mathcal{A}$ is the action taken from a *behavior policy*, and $r_i$ is the observed reward given by $r_i = r(s_i, a_i) + \varepsilon_i$, where $\varepsilon_t$ is 1-sub-Gaussian (Lattimore & Szepesvári, 2020, Definition 5.2). In this work, we consider the *KL-regularized objective*

$$J_\eta(\pi) := \mathbb{E}_{(s,a)\sim\rho\times\pi}\left[r(s,a) - \eta^{-1}\log\frac{\pi(a|s)}{\pi^{\mathsf{ref}}(a|s)}\right], \tag{2.1}$$

where $\pi^{\mathsf{ref}}$ is a known reference policy and the "inverse temperature" $\eta$ controls the intensity of regularization. For simplicity, we assume that $\pi^{\mathsf{ref}}$ is also the behavior policy that generates the dataset $\mathcal{D}$, which is similar to the type of "behavior regularization" studied in Zhan et al. (2022). The unique optimal policy $\pi_\eta^* := \mathrm{argmax}_{\pi\in\Delta(\mathcal{A}|\mathcal{S})} J_\eta(\pi)$ is given by (See, e.g., Zhang 2023, Proposition 7.16)[3]

$$\pi^*(\cdot|s) \propto \pi^{\mathsf{ref}}(\cdot|s)\exp\left(\eta\cdot r(s,\cdot)\right), \forall s\in\mathcal{S}. \tag{2.2}$$

A policy $\pi$ is said to be $\epsilon$-optimal if $\mathrm{SubOpt}_{\mathrm{RKL}}(\pi) := J(\pi^*) - J(\pi) \le \epsilon$ and the goal of the agent is to find one such policy using $\mathcal{D}$. Note that $\mathrm{SubOpt}_{\mathrm{RKL}}(\cdot)$ is defined through (2.1) and thus **depends on** $\eta$. To ensure that $\epsilon$-optimality is achievable, we assume that $r$ lies in a known function class $\mathcal{G} \subset (\mathcal{S}\times\mathcal{A} \to [0,1])$, from which the agent obtains an estimator $\widehat{r}$. More specifically, we work with general function approximation under realizability, which is as follows.

**Assumption 2.1.** For this known function class $\mathcal{G} \subset (\mathcal{S}\times\mathcal{A} \to [0,1])$, $\exists g^* \in \mathcal{G}$ with $g^* = r$.

We also employ the standard notion of covering number (Wainwright, 2019, Definition 5.1) as the complexity measure of the reward function class $\mathcal{G}$.

**Definition 2.2** ($\epsilon$-net and covering number). Given a function class $\mathcal{G} \subset (\mathcal{S}\times\mathcal{A}\to\mathbb{R})$, a finite set $\mathcal{G}(\epsilon) \subset \mathcal{G}$ is an $\epsilon$-net of $\mathcal{G}$ w.r.t. $\|\cdot\|_\infty$, if for any $g\in\mathcal{G}$, there exists $g'\in\mathcal{G}(\epsilon)$ such that $\|g-g'\|_\infty \le \epsilon$. The $\epsilon$-covering number is the smallest cardinality $\mathcal{N}_\mathcal{G}(\epsilon)$ of such $\mathcal{G}(\epsilon)$.

**Assumption 2.3.** For any $\epsilon_c > 0$, the $\epsilon_c$-covering number $\mathcal{N}_\mathcal{G}(\epsilon_c)$ of $\mathcal{G}$ is $\mathrm{poly}(\epsilon_c^{-1})$.

Assumption 2.3 allowing $\log\mathcal{N}_\mathcal{G}(\epsilon)$ to be roughly negligible is arguably mild. For example, when $\mathcal{G}$ is the class of linear functions of dimension $d$ and radius $R$, the covering number is $\mathcal{N}_\mathcal{G}(\epsilon) = O((1 + R\epsilon^{-1})^d)$ (Jin et al., 2020, Lemma D.6), which satisfies Assumption 2.3.

**Concentrability.** The data quality of $\mathcal{D}$ collected by $\pi^{\mathsf{ref}}$ is typically characterized by *concentrability* in offline RL (Farahmand et al., 2010; Chen & Jiang, 2019; Jiang & Xie, 2024), which quantifies the ability of the behavioral policy to generate diverse actions. We first define the density-ratio-based concentrability as follows.

**Definition 2.4** (*Density-ratio-based* concentrability). For policy class $\Pi$, reference policy $\pi^{\mathsf{ref}}$, the density-ratio-based all-policy concentrability $C^\Pi$ is $C^\Pi := \sup_{\pi\in\Pi,s\in\mathcal{S},a\in\mathcal{A}}\pi(a|s)/\pi^{\mathsf{ref}}(a|s)$, whose single-policy counterpart under the optimal policy $\pi^*$ is $C^{\pi^*} := \sup_{s\in\mathcal{S},a\in\mathcal{A}}\pi^*(a|s)/\pi^{\mathsf{ref}}(a|s)$.

---

[3]We suppress $J_\eta$ into $J$ and $\pi_\eta^*$ into $\pi^*$ when they are clear in context in the following presentation.

In the definition above, small all-policy concentrability intuitively corresponds to $\mathrm{supp}(\pi^{\mathsf{ref}})$ covering all possible inputs. On the other hand, small single-policy concentrability means that $\mathrm{supp}(\pi^{\mathsf{ref}})$ only subsumes $\mathrm{supp}(\pi^*)$. In this paper, in addition to density-ratio-based concentrability, we also adopt the following $D^2$-based concentrabilites to better capturing the nature of function class $\mathcal{G}$. In detail, we start with the $D^2$-divergence as follows.

**Definition 2.5.** Given a function class $\mathcal{G} \subset (\mathcal{S} \times \mathcal{A} \to \mathbb{R})$ and a fixed policy $\pi$, define the $D^2$-divergence $D^2_{\mathcal{G}}((s, a); \pi)$ as

$$\sup_{g, h \in \mathcal{G}} \frac{\big(g(s, a) - h(s, a)\big)^2}{\mathbb{E}_{(s', a') \sim \rho \times \pi}[(g(s', a') - h(s', a'))^2]}.$$

The "eluder dimension"-type Definition 2.5 is directly inspired by Di et al. (2024); Zhao et al. (2024), the intuition behind which is that given $(s, a) \in \mathcal{S} \times \mathcal{A}$, a small $D^2$-divergence indicates that for two functions $g$ and $h$, if they are close under the behavior policy $\pi$, then they will also be close on such pair $(s, a)$. Therefore, the $D^2$-divergence quantifies how well the estimation on dataset collected by the behavior policy $\pi$ can be generalized to a specific state-action pair.

**Remark 2.6.** For the tabular setting, a direct computation yields $D^2(s, a) = (\rho(s)\pi^{\mathsf{ref}}(a|s))^{-1}$, which can be estimated by the visitation frequency empirically. Under linear function approximation, it is well known that $D^2(s, a) = \|\phi(s, a)\|^2_{\Sigma^{-1}}$ under mild conditions of the parameter space, where $\Sigma = \mathbb{E}_{\rho \times \pi^{\mathsf{ref}}}\phi(s, a)\phi(s, a)^\top$ is the covariance matrix, which can be estimated by empirical covariance matrices in practice, potentially with ridge regularization. For more general function classes like neural networks, the $D^2$ can also be efficiently approximated by heuristics as discussed in Xiong et al. (2024); Gupta et al. (2024); Xu et al. (2025).

We are now ready to define the two notions of concentrability conditions.

**Assumption 2.7** (All-policy concentrability). Given a reference policy $\pi^{\mathsf{ref}}$, there exists $D < \infty$ such that $D^2 = \sup_{(s,a) \in \mathcal{S} \times \mathcal{A}} D^2_{\mathcal{G}}((s, a); \pi^{\mathsf{ref}})$.

Assumption 2.7 indicates that the errors on any state-action pairs can be bounded by the error on the samples from $\rho \times \pi$ up to a factor $D$, whose relaxed counterpart under the same $\pi^{\mathsf{ref}}$ is as follows.

**Assumption 2.8** (Single-policy concentrability). $D^2_{\pi^*} := \mathbb{E}_{(s,a) \sim \rho \times \pi^*} D^2_{\mathcal{G}}((s, a); \pi^{\mathsf{ref}}) < \infty$.

Assumption 2.8 indicates that the errors on the distributions of state-action pairs $\rho \times \pi^*$ can be bounded by the error on the samples from $\rho \times \pi^{\mathsf{ref}}$ up to some constant. For both types, the single-policy concentrability assumption is strictly weaker than the all-policy concentrability assumption. However, in general, the two quantities characterizing single-policy concentrability $C^{\pi^*}$ and $D^2_{\pi^*}$ cannot be bounded by each other up to constant factors. In particular, we have $D^2_{\pi^*} \le |\mathcal{S}||\mathcal{A}|C^{\pi^*}$, indicating that $C^{\pi^*}$ subsumes $D^2_{\pi^*}$ when $|\mathcal{S}|$ and $|\mathcal{A}|$ can be seen as constants. We refer the reader to Appendix B for a further discussion on the relation between $C^{\pi^*}$ and $D^2_{\pi^*}$.

## 2.2 ALGORITHM

In this subsection, we present an offline bandit algorithm, KL-PCB, for KL-regularized contextual bandits in Algorithm 1. KL-PCB first leverages least-square estimator to find a function $\bar{g} \in \mathcal{G}$ that minimizes its risk on the offline dataset. In Zhao et al. (2024), such $\bar{g}$ is directly applied to construct the estimated policy. In contrast, we construct a pessimistic estimator of $g^*$ following the well-known pessimism principle in offline RL (Jin et al., 2021). Specifically, we define the bonus term $\Gamma_n$ through the confidence radius $\beta = \sqrt{128 \log\big(2\mathcal{N}_{\mathcal{G}}(\epsilon)/\delta\big)/3n + 18\epsilon}$ as

$$\Gamma_n(s, a) = \beta D_{\mathcal{G}}\big((s, a), \pi^{\mathsf{ref}}\big), \forall (s, a) \in \mathcal{S} \times \mathcal{A}. \tag{2.3}$$

We then obtain our pessimistic estimation $\widehat{g}$ by setting $\widehat{g} = \bar{g} - \Gamma_n$, which is less than $g^*$ with high probability. Formally, let the event $\mathcal{E}(\delta)$ given $\delta > 0$ defined as

$$\mathcal{E}(\delta) := \Big\{ \sup_{(s,a) \in \mathcal{S} \times \mathcal{A}} \big[|\bar{g} - g^*| - \Gamma_n\big](s, a) \le 0 \Big\}, \tag{2.4}$$

on which the least square estimation $\bar{g}$ obtained in Line 1 of Algorithm 1 does not deviate too much from the true function $g^*$ and therefore $\widehat{g}$ is a pessimistic estimation of $g^*$. We have the following lemma indicating that this event holds with high probability.

---

**Algorithm 1** Offline KL-Regularized Pessimistic Contextual Bandits (KL-PCB)

---

**Require:** regularization $\eta$, reference policy $\pi^{\mathsf{ref}}$, offline dataset $\mathcal{D}$, function class $\mathcal{G}$
 1: Least square estimation of reward function $\bar{g} \in \operatorname{argmin}_{g \in \mathcal{G}} \sum_{(s_i,a_i,r_i) \in \mathcal{D}} \big(g(s_i, a_i) - r_i\big)^2$
 2: Let $\widehat{g} \leftarrow \bar{g} - \Gamma_n$, where $\Gamma_n$ is the bonus term in (2.3)
**Ensure:** $\widehat{\pi}(a|s) \propto \pi^{\mathsf{ref}}(a|s) \exp\big(\eta \cdot \widehat{g}(s,a)\big)$

---

**Lemma 2.9.** For all $\delta > 0$, $\mathcal{E}(\delta)$ holds with probability at least $1 - \delta$.

After obtaining the pessimistic estimation, KL-PCB output the policy $\widehat{\pi}$, which maximizes the estimated objective

$$\widehat{J}(\pi) = \mathbb{E}_{(s,a)\sim\rho\times\pi}\left[\widehat{g}(s,a) - \eta^{-1}\log\frac{\pi(a|s)}{\pi^{\mathsf{ref}}(a|s)}\right],$$

the maximizer of which is the counterpart of (2.2), i.e.,

$$\widehat{\pi}(a|s) \propto \pi^{\mathsf{ref}}(a|s)\exp\big(\eta \cdot \widehat{g}(s,a)\big).$$

## 2.3 Theoretical Results

The sample complexity for KL-regularized contextual bandits is settled in this subsection. We first give the upper bound of KL-PCB.

**Theorem 2.10.** Under Assumption 2.8, for sufficiently small $\epsilon \in (0,1)$, if we set $\Gamma_n$ as in (2.3), then $n = \widetilde{O}\big(\eta D_{\pi^*}^2 \epsilon^{-1}\log\mathcal{N}_{\mathcal{G}}(\epsilon)\big)$ suffices to guarantee the output policy $\widehat{\pi}$ of Algorithm 1 to be $\epsilon$-optimal with probability at least $1 - \delta$.

Previously, Zhao et al. (2024) achieved an $\widetilde{O}(\epsilon^{-1})$ sample complexity under Assumption 2.7. As a comparison, KL-PCB achieves the same $\widetilde{O}(\epsilon^{-1})$ sample complexity but only requiring Assumption 2.8, which is weaker than Assumption 2.7. We also provide the sample complexity lower bound of KL-regularized contextual bandits in the following theorem, which, together with Theorem 2.10, demonstrates that single-policy concentrability is both necessary and sufficient for near-optimal offline learning evaluated by KL-regularized objectives.

**Theorem 2.11.** For $\forall S \geq 1$, $\eta > 4\log 2$, $C^* \in (2, \exp(\eta/4)]$, and any algorithm $\mathsf{Alg}$, there is a KL-regularized contextual bandit with $C^{\pi^*} \leq C^*$ such that $\mathsf{Alg}$ requires $\Omega\big(\min\{\eta\epsilon^{-1}, \epsilon^{-2}\}C^*\log\mathcal{N}_{\mathcal{G}}(\epsilon)\big)$ samples to find an $\epsilon$-optimal policy for sufficiently small $\epsilon$.

Previously, Zhao et al. (2024) provided a sample complexity lower bound of $\Omega(\eta\log\mathcal{N}_{\mathcal{G}}(\epsilon)/\epsilon)$ under KL regularization. Foster et al. (2025) also provided a lower bound of $\Omega(C^{\pi^*})$ for KL-regularized objective to show the necessity of coverage. Compared to their results, our result shows that the *multiplicative* dependency on $C^{\pi^*}$ is necessary for the first time.

**Remark 2.12.** Theorem 2.11 shows that when $\epsilon$ is sufficiently small, any algorithm for offline KL-regularized contextual bandits requires at least $\Omega(\eta C^{\pi^*})\epsilon^{-1}\log\mathcal{N}_{\mathcal{G}}(\epsilon)$ samples to output an $\epsilon$-optimal policy. The presence of $\exp(\mathrm{poly}(\eta))$ in the range of $C^*$ is inevitable, since we always have $C^{\pi^*} \leq \exp(\eta)$ in reverse KL regularized bandits with bounded rewards.

**Remark 2.13.** As discussed before, we might have some easy instances with $D_{\pi^*}^2 \leq C^{\pi^*}$, where KL-PCB outperforms the lower bound. This does not volates Theorem 2.11 since Theorem 2.11 only guarantees that *there exist* some hard instances that all algorithms require at least $\Omega(\min\{\eta\epsilon^{-1}, \epsilon^{-2}\}C^*\log\mathcal{N}_G(\epsilon))$ samples.

## 2.4 Proof Overview of Theorem 2.10

In this section, we summarize the novel techniques in the proof of Theorem 2.10, which is deferred to Appendix D.4. At a high level, if we consider the regularized objective (1.1) multi-arm bandits, then $P \mapsto \mathsf{KL}\,(P\|Q)$ is 1-strongly convex w.r.t. $\mathsf{TV}\,(\cdot\|\cdot)$ (Polyanskiy & Wu, 2025, Exercise I.37), and thus $J(\pi)$ is strongly concave. Therefore, $J(\pi^*) - J(\widehat{\pi})$ is possible to be of the order $[\mathsf{TV}\,(\pi^*\|\widehat{\pi})]^2 \approx \widetilde{O}(n^{-1})$, pretending that $\pi^*$ is the unconstrained maximizer. In detail, we follow the regret decomposition in Zhao et al. (2024), which is encompassed by the following lemma.

**Lemma 2.14.** Let $g : \mathcal{S} \times \mathcal{A} \to \mathbb{R}$ be any reward function, then there exist some $\gamma \in [0,1]$ such that the sub-optimality gap of $\pi_g(\cdot|s) \propto \pi^{\text{ref}}(\cdot|s) \exp\big(\eta g(s, \cdot)\big)$ can be bounded as

$$J(\pi^*) - J(\pi_g) \leq \eta \mathbb{E}_{(s,a)\sim\rho\times\pi_\gamma}\big[(g^* - g)^2(s,a)\big],$$

where $g_\gamma \coloneqq \gamma g + (1-\gamma)g^*$ and $\pi_\gamma(\cdot|s) \propto \pi^{\text{ref}}(\cdot|s) \exp\big(\eta g_\gamma(s, \cdot)\big)$.

In Zhao et al. (2024), because the $g$ in Lemma 2.14 is substituted with only the least-square estimator $\bar{g}$ with no extra structures, the reliance on the "mid-point" policy $\pi_\gamma$ can only be controlled all-policy concentrability. However, our $g$ is the pessimistic estimator $\widehat{g}$ of $g^*$ in Algorithm 1, and thus the presence of $\pi_\gamma$ can be eliminated for free: let $G(\gamma) \coloneqq \mathbb{E}_{\rho\times\pi_\gamma}\Big[\big(\widehat{g} - g^*\big)^2(s,a)\Big]$ and $\triangle(s,a) \coloneqq \big(\widehat{g} - g^*\big)(s,a) \leq 0$, then a direct computation (detailed in the proof of Lemma D.3) yields

$$G'(\gamma) = \eta\mathbb{E}_\rho\Big[\mathbb{E}_{\pi_\gamma}\big[\triangle^3(s,a)\big] - \mathbb{E}_{\pi_\gamma}\big[\triangle^2(s,a)\big]\mathbb{E}_{\pi_\gamma}\big[\triangle(s,a)\big]\Big] \leq 0. \tag{2.5}$$

This gives $J(\pi^*) - J(\widehat{\pi}) \leq \eta\mathbb{E}_{\rho\times\pi^*}\big[(\widehat{g} - g^*)^2(s,a)\big]$, which can be bounded with single-policy concentrability while still achieves the sharp dependency $\epsilon^{-1}$ on $\epsilon$. Here, (2.5) holds due to a moment-based machinery in Lemma 2.15.

**Lemma 2.15.** If $\mathbb{P}(X \leq 0) = 1$ and $\mathbb{E}|X|^3 < \infty$, then $\mathbb{E}[X^3] - \mathbb{E}[X^2]\mathbb{E}[X] \leq 0$.

The intuition behind Lemma 2.15 is natural: $X$ and $X^2$ cannot be positively correlated. Moreover, to the best of our knowledge, we are the first to unveil this moment-based structure in our non-standard pessimism-based analysis, from which the sharp upper bound follows. While pessimism is widely adopted to derive near-optimal statistical rates under single-policy concentrability in offline RL with reward maximization as the goal (See, e.g., Jin et al. (2021); Xiong et al. (2022)), the standard pessimism-based pipeline is not sharp enough for bounding the $\text{SubOpt}_{\text{RKL}}(\widehat{\pi})$ *defined through regularized objectives*, the reason of which is detailed in the last paragraph of Appendix A.1.

## 3 $f$-DIVERGENCE-REGULARIZED CONTEXTUAL BANDITS

As discussed in Section 2, the fast rate implied by Theorems 2.10 and 2.11 is primarily achieved due to the strong convexity of $\pi \mapsto \text{KL}(\pi\|\pi^{\text{ref}})$. However, KL is just an instance of $f$-divergence with $f(x) = x\log x$, which is only locally strongly convex but not strongly convex. Motivated by this observation, we further examine $f$-divergence regularization with strongly convex $f$, which may introduce a more favorable curvature in the performance metric of offline learning in principle.

### 3.1 PROBLEM SETUP

We study a contextual bandit setting similar to that in Section 2.1. In this section, we consider the following $f$-divergence regularized objective

$$J_{\eta,D_f}(\pi) \coloneqq \mathbb{E}_{(s,a)\sim\rho\times\pi}[r(s,a)] - \eta^{-1}\mathbb{E}_{s\sim\rho}\big[D_f\big(\pi(\cdot|s)\|\pi^{\text{ref}}(\cdot|s)\big)\big], \tag{3.1}$$

where $\eta$ is the regularization intensity and $D_f(p\|q) \coloneqq \mathbb{E}_{a\sim q}\Big[f\big(p(a)/q(a)\big)\Big]$ is the $f$-divergence. Let the optimal policy be $\pi^*_{\eta,D_f} \coloneqq \text{argmax}_{\pi\in\Delta(\mathcal{A}|\mathcal{S})} J_{\eta,D_f}(\pi)$ and we re-define the learning objective as searching for a policy $\pi$ with $\text{SubOpt}_{f\text{div}}(\pi) \coloneqq J(\pi^*) - J(\pi) \leq \epsilon$.[4] We consider those functions $f : (0, +\infty) \to \mathbb{R}$ with a nice positive curvature condition in Assumption 3.1.

**Assumption 3.1.** $f$ is $\alpha$-strongly convex, twice continuously differentiable, and $f(1) = 0$.

Many elementary functions like quadratic polynomials naturally satisfy Assumption 3.1. For instance, the 1-strongly convex $f(x) = (x - 1)^2/2$ induces $D_f(P\|Q) = \chi^2(P\|Q)$, which is the $\chi^2$-divergence recently considered in RL literature (see e.g., Zhan et al. (2022); Huang et al. (2025b); Amortila et al. (2024)). This regularization exhibits a promising theoretical potential for relaxing the data coverage requirement for efficient offline policy learning (Huang et al., 2025b) and to be effective in preventing reward hacking (Laidlaw et al., 2025) against unregularized objectives. These favorable benefits are primary due to the observation that strongly convex $f$'s impose a stronger penalization on actions out of the coverage of $\pi^{\text{ref}}$.

---

**Algorithm 2** Offline $f$-divergence Regularized Contextual Bandits ($f$-CB)

---

**Require:** regularization $\eta$, reference policy $\pi^{\text{ref}}$, function class $\mathcal{G}$, offline dataset $\mathcal{D}$

1: Least square estimation $\bar{g} \in \operatorname{argmin}_{g \in \mathcal{G}} \sum_{(s_i, a_i, r_i) \in \mathcal{D}} \left(g(s_i, a_i) - r_i\right)^2$

2: Compute the optimal policy under the least-square reward estimator $\bar{g}$ for $s \in \mathcal{S}$ as

$$\widehat{\pi}(\cdot|s) \leftarrow \operatorname*{argmax}_{\pi(\cdot|s) \in \Delta(\mathcal{A})} \langle \pi(\cdot|s), \bar{g}(s, \cdot) \rangle + \eta^{-1} D_f\left(\pi(\cdot|s) \| \pi^{\text{ref}}(\cdot|s)\right)$$

**Ensure:** $\widehat{\pi}$

---

## 3.2 ALGORITHM AND MAIN RESULTS

In this subsection, we present an offline learning algorithm for $f$-divergence regularized bandit, $f$-CB, in Algorithm 2. Algorithm 2 first leverages least-square estimator to find a function $\bar{g} \in \mathcal{G}$ that minimizes its risk on the offline dataset. The algorithm then uses the least squares estimation $\bar{g}$ to construct the output policy $\widehat{\pi}$. Compared to Algorithm 1, $f$-CB does not require any procedure to construct pessimistic reward estimation, whose sample complexity upper bound is given as follows.

**Theorem 3.2.** Under Assumption 3.1, for sufficiently small $\epsilon \in (0, 1)$, with probability at least $1 - \delta$, $n = \widetilde{O}(\alpha^{-1} \eta \epsilon^{-1} \log \mathcal{N}_\mathcal{G}(\epsilon))$ is sufficient to guarantee the output policy $\widehat{\pi}$ of $f$-CB to be $\epsilon$-optimal.

**Remark 3.3.** Compared to the $D^2_{\pi^*}$ dependency in Theorem 2.10, Theorem 3.2 shows that the sample complexity of Algorithm 2 gets rid of the dependency on any data coverage conditions when $f$ is strongly convex. Intuitively, this is because the $f$-divergence regularization in this case is much stronger, so that both $\pi^*$ and $\widehat{\pi}$ are close enough to $\pi^{\text{ref}}$.

The following hardness result justify the near-optimality of Theorem 3.2 for $f$-divergence-regularized contextual bandits.

**Theorem 3.4.** For any $\epsilon \in (0, 1), \alpha > 0, \eta > 0, S > 32/3 \cdot \log 2$, sufficiently small $\epsilon$, and algorithm Alg, there is an $\alpha$-strongly-convex function $f$ and an $f$-divergence-regularized contextual bandit instance such that Alg requires at least $\Omega\left(\alpha^{-1} \eta \epsilon^{-1} \log \mathcal{N}_\mathcal{G}(\epsilon)\right)$ samples to return an $\epsilon$-optimal policy.

## 3.3 PROOF OVERVIEW OF THEOREM 3.2

We provide an overview of key analysis techniques for proving Theorem 3.2. Unlike KL-regularization, the $\pi^*$ under $f$-divergence might not have a closed form. This means that the proof of Lemma 2.14, which relies on the closed form of $\pi^*$, cannot be directly adopted. Therefore, we address this from a dual-Bregman perspective, inspired by Abernethy et al. (2015). For the simplicity of presentation, we consider multi-armed bandits here and omit the subscript for context $s$.

We consider the function $H(\pi) = \eta^{-1} D_f(\pi \| \pi^{\text{ref}})$, which is the regularizer in the objective. Then its convex conjugate is given by $H^*(r) = \sup_{\pi \in \Delta^d}\{\langle \pi, r \rangle - H(\pi)\}$, which is exactly the expected reward obtained by the optimal policy given reward function $r$. One observation is that when $f$ is strongly convex, the induced $f$-divergence, and therefore the function $H$ are also strongly convex. Therefore, let $\pi_r = \operatorname{argmax}_\pi\{\langle \pi, r \rangle - H_s(\pi)\}$ given some reward function $r$, the strong convexity of $H(\pi)$ gives that $\nabla H^*(r) = \pi_r$. This leads to the following regret decomposition, which is one of our key observations:

$$J(\pi^*) - J(\widehat{\pi}) = \mathbb{E}_{a \sim \pi^*}[g^*(a)] - \mathbb{E}_{a \sim \widehat{\pi}}[g^*(a)] - \eta^{-1}\left[D_f(\pi^* \| \pi^{\text{ref}}) - D_f(\widehat{\pi} \| \pi^{\text{ref}})\right]$$
$$= H^*(g^*) - H^*(\bar{g}) - \langle \widehat{\pi}, g^* - \bar{g} \rangle$$
$$= H^*(g^*) - H^*(\bar{g}) - \langle \nabla H^*(\bar{g}), g^* - \bar{g} \rangle,$$

which is the Bregman divergence of the dual function $H^*$ and therefore can be bounded by $(g^* - \bar{g})^\top \nabla^2 H^*(\widetilde{g})(g^* - \bar{g})$ for some $\widetilde{g}$. By Proposition 3.2 in Penot (1994), when $H$ is strongly convex, we can bound $\nabla^2 H^*(\widetilde{g})$ as follows

$$\nabla^2 H^*(\widetilde{g}) \preceq \left(\nabla^2 H(\pi_{\widetilde{g}})\right)^{-1} \preceq \alpha^{-1} \eta \operatorname{diag}\left(\pi^{\text{ref}}(a_1), \cdots, \pi^{\text{ref}}(a_{|\mathcal{A}|})\right),$$

---

[4]We again suppress $J_{\eta, D_f}(\cdot)$ into $J(\cdot)$ and $\pi^*_{\eta, D_f}$ into $\pi^*$ when there is no confusion.

which enables us to bound $(g^* - \bar{g})^\top \nabla^2 H^*(\tilde{g})(g^* - \bar{g})$ by $\alpha^{-1}\eta\mathbb{E}_{\pi^{\mathrm{ref}}}[(g^* - \hat{g})^2]$. Since $\mathbb{E}_{\pi^{\mathrm{ref}}}[(g^* - \hat{g})^2]$ is not related to $\pi^*$, the upper bound is independent of any notion of concentrability.

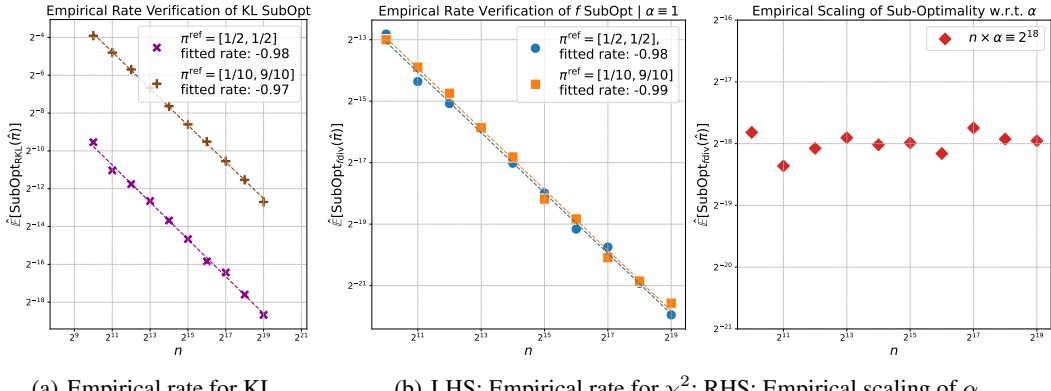

Figure 1: The empirical relation between $\log_2 n$ and $\log_2 \mathrm{SubOpt}$. The *fitted rate* means the slope of $\log_2 n \sim \log_2 \mathrm{SubOpt}$ estimated via linear regression. Here $n$ is the sample size. Every point is the average over **100** independent trials.

## 4 EXPERIMENTS

**Simulation on multi-armed bandits.** We first empirically check the correctness of our matching bounds for KL and $f$-divergence on the simplest testbed: *two-armed* bandits, i.e., $\mathcal{A} = \{0, 1\}$. We use one hard instance constructed in the proof of Theorem 2.11 (Appendix D.5) for the simulation under KL and one hard instance constructed in the proof of Theorem 3.4 (Appendix E.2) for the simulation under $f$-divergence with $f(x) = \alpha(x - 1)^2/2$. Recall that the dependency on $\epsilon$ in all sample complexity bounds above is $\widetilde{\Theta}(\epsilon^{-1})$, and thus both $\mathrm{SubOpt}_{\mathrm{RKL}}$ and $\mathrm{SubOpt}_{f\mathrm{div}}$ should be roughly proportional to $n^{-1}$ as a function of the sample size $n$, which can be verified from the linear regression between $\log_2 n$ and $\log_2 \mathrm{SubOpt}$; i.e., the estimated slope should be approximately $-1$. Therefore, the two fitted rates in Figure 1(a) indicates that KL-PCB indeed achieves the near-optimal statistical rate $n^{-1}$ under different $\pi^{\mathrm{ref}}$'s and the counterparts in the LHS of Figure 1(b) indicates the near-optimality of $f$-CB empirically. The contrast between Figure 1(a) and the LHS of Figure 1(b) also corroborates that the sample complexity against the KL-regularized objective positively depend on the concentrability, while that against the $\chi^2$-divergence-regularized objective does not vary with the coverage condition of $\pi^{\mathrm{ref}}$. Moreover, on top of the hard instance for $f$-divergence, we further set $\alpha = 2^{15}/n$ to numerically examine the scaling of $\mathrm{SubOpt}_{f\mathrm{div}}$ w.r.t. the strong convexity modulus $\alpha$. As shown on the RHS of Figure 1(b), $\mathrm{SubOpt}_{f\mathrm{div}}$ remains stable as $n$ goes up given $n\alpha \equiv 2^{15}$; therefore, Figure 1(b) also empirically verified that $\mathrm{SubOpt}_{f\mathrm{div}}$ is inversely proportional to $\alpha$.

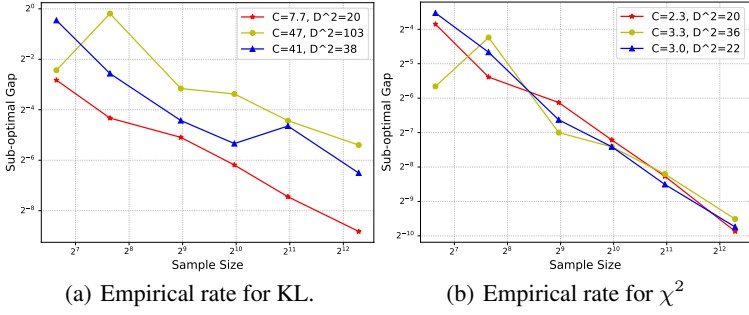

Figure 2: The empirical relation between $\log_2 n$ and $\log_2 \mathrm{SubOpt}$ for linear bandits. In the legend, we denote $C^{\pi^*}$ (resp. $D^2_{\pi^*}$) by `C` (resp. `D^2`).

**Simulation on linear bandits.** We then simulate a linear bandit as follows. The constructions of the feature map $\phi$, ground-truth parameter $\boldsymbol{\theta}^*$ and the induced reward are detailed in Appendix C.1. The behavior policy is constructed as $\pi^{\mathsf{ref}} = \beta\mathsf{Unif}(\mathcal{A}) + (1-\beta)\mathsf{Unif}(\mathcal{A}_k)$, where $\mathcal{A}_k \subset \mathcal{A}$ be the subset such that $\mathcal{A}_k$ consists of the $k$ arms with the lowest expected reward. We consider three different behavior policies, $(\beta, k) \in \{(1, \cdot), (0.1, 4), (0.05, 20)\}$, which induces various $C^{\pi^*}$ and $D_{\pi^*}^2$ so as to demonstrate the influence of coverage under different regularization. The results are compiled in Figure 2. Specifically, for the KL-regularized cases depicted in Figure 2(a), we see that as the coverage coefficients $C^{\pi^*}$ and $D_{\pi^*}^2$ vary, there is a consistent sub-optimality gap margin between these instances. In contrast, Figure 2(b)shows that the sub-optimality gaps under different instances (with distinct coverage coefficient) are very close for sufficiently large sample sizes. These results corroborate our theoretical finding that the sample complexity w.r.t. KL-regularized objectives is concentrability-dependent but that w.r.t $f$-divergence ones is not (for strongly convex $f$).

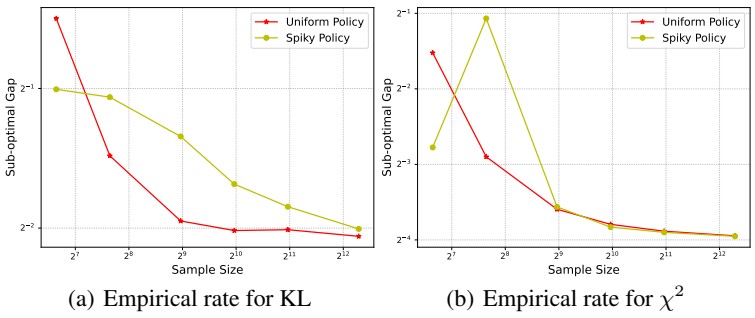

(a) Empirical rate for KL        (b) Empirical rate for $\chi^2$

Figure 3: The empirical relation between $\log_2 n$ and $\log_2 \mathrm{SubOpt}$ on MNIST dataset.

**Real-world experiments.** We further verify our theory on a vision dataset, MNIST (LeCun, 1998). The construction of the feature map is detailed in Appendix C.2. We consider two reference polices, a uniform policy $\mathsf{Unif}(\mathcal{A})$ and a spiky policy $0.5\mathsf{Unif}(\mathcal{A}) + 0.5\mathsf{Dirac}(\{0\})$ to obtain instances with different concentrability coefficients. Figure 3 exhibits the $\mathrm{SubOpt}$ curves, which show that under KL-regularization, when sample size is not large enough, there exists a considerable gap between instances with different behavior policy, but the gap is vanishing as the sample size increases. On the other hand, as for $\chi^2$-divergence regularization, such a gap vanishes quickly when the sample size becomes moderately large and the sub-optimal gap remains similar for larger sample sizes. These results are consistent with the simulation in Section 4 and our theoretical findings.

## 5 CONCLUSION AND FUTURE WORK

In this work, we take the first step towards fully understanding the statistical efficiency *with respect to $f$-divergence-regularized objectives* of offline policy learning by sharp analyses for two empirically relevant subclasses. (1) We are the first to show that single-policy concentrability is nearly the right coverage condition for reverse KL to achieve the fast $\widetilde{\Theta}(\epsilon^{-1})$ sample complexity. The novel techniques in algorithm analysis leverages the curvature of KL-regularized objectives and integrates pessimism with a newly identified moment-based observation, enabling a neat refinement of a mean-value-type argument to the extreme; which are decoupled from tricky algorithmic tweaks, and thus might be of independent interest. (2) If strong convexity is further imposed on $f$, our fast $\widetilde{\Theta}(\epsilon^{-1})$ sample complexity is provably free of any coverage dependency. Unlike those for KL, the upper bound arguments for strongly convex $f$ do not rely on specific closed-form solutions of the regularized objective maximizer.

All techniques in this work can be generalized beyond vanilla absolute reward feedback, as certified by CDBs, which is detailed in Appendix F under a slightly different notion of $D^2$ tailored for pairwise comparison feedback. However, for reverse-KL regularization, the $D_{\pi^*}^2$ in the upper bound and the $C^{\pi^*}$ in the lower bound still does not perfectly match. Also, for general $f$-divergence other than reverse-KL, our analyses require $f$ to be twice-continuously differentiable and strongly convex. Fully closing the gap under reverse-KL regularization and extending the analysis to general $f$-divergences are interesting directions for future work.

## THE USE OF LARGE LANGUAGE MODELS (LLMS)

We use LLMs as a tool to refine our writing and correct grammatical errors.

## ACKNOWLEDGMENT

We thank the anonymous reviewers and area chair for their helpful comments. QZ, KJ, HZ and QG are supported in part by the National Science Foundation DMS-2323113 and IIS-2403400. HZ is also supported in part by Amazon PhD Fellowship. The views and conclusions contained in this paper are those of the authors and should not be interpreted as representing any funding agencies.

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

## CONTENTS

## H   Auxiliary Lemmas                                                            35

## A   ADDITIONAL REVIEW OF EXISTING RESULTS

**Additional notations.**   Besides the notation introduced in Section 1, we will use the following notations in Appendix. We denote $[N] := \{1, \cdots, N\}$ for any positive integer $N$. Boldfaced lower case (resp. upper case) letters are reserved for vectors (resp. matrices). Given a positive definite $\mathbf{\Sigma} \in \mathbb{R}^{d \times d}$ and $\mathbf{x} \in \mathbb{R}^d$, we denote the vector's Euclidean norm by $\|\mathbf{x}\|_2$ and define $\|\mathbf{x}\|_{\mathbf{\Sigma}} = \sqrt{\mathbf{x}^\top \mathbf{\Sigma} \mathbf{x}}$. We use $\mathsf{Bern}(p)$ to denote Bernoulli distribution with expectation $p$ and $\mathsf{Unif}(\mathcal{X})$ for the uniform distribution on finite set $\mathcal{X}$. For $x \in \mathbb{R}^{|\mathcal{A}|}$, we denote $\|x\|_1 = \sum_{a \in \mathcal{A}} |x_a|$. We also denote $x_n = \Omega(y_n)$ by $x_n \gtrsim y_n$ in Appendix. We use $d_H$ for Hamming distance.

### A.1   PREVIOUS ATTEMPTS ON UNDERSTANDING KL-REGULARIZED RL

There has been a surge of interest in understanding the principle behind KL-regularized RL. Ahmed et al. (2019); Liu et al. (2019) studied by ablation the effect of entropy regularization on the stability of policy improvement in policy optimization, the regret of which has been rigorously settled under the classic online mirror descent framework (Cai et al., 2020; He et al., 2022; Ji et al., 2023). Neu et al. (2017) unified popular KL-regularized policy optimization algorithms under a convex optimization framework, but the interplay with the data was left untouched. A series of work (Geist et al., 2019; Vieillard et al., 2020; Kozuno et al., 2022) then analyzed the sample complexity of algorithms using KL/entropy-type proximal terms with respect to the previous iteration or/and entropy regularizer with improved dependence on the effective horizon in discounted Markov decision processes. However, the performance metric in these studies is still the unregularized reward maximization objective, under which the sample complexity for finding an $\epsilon$-optimal policy is at least equal to the statistical limit $\Omega(\epsilon^{-2})$.

**Convergence under regularized objectives.**   Several recent studies (Xie et al., 2024; Xiong et al., 2024; Zhao et al., 2024; 2025; Foster et al., 2025) switched the focus to analyzing the sub-optimality guarantee with respect to the regularized objective (1.1). In particular, Xie et al. (2024) studied token-level Markov decision processes (MDPs) and proposed a KL-regularized RL algorithm named XPO, which achieves $\widetilde{O}(\epsilon^{-2})$ sample complexity under their notion of all-policy concentrability. Xiong et al. (2024) proposed an Offline GSHF algorithm via the principle of *pessimism in the face of uncertainty*, and proved $\widetilde{O}(\epsilon^{-2})$ sample complexity under single-policy concentrability (See Section 2.1 for detailed definitions of concentrability). On the other hand, the sharp analysis in Zhao et al. (2024) yields the optimal sample complexity $\widetilde{O}(\epsilon^{-1})$, but requires all-policy concentrability (Zhao et al., 2024, Definition 2.6), i.e., the behavior policy $\pi^{\mathsf{ref}}$ is required to cover the entire function class for all possible policies. Zhao et al. (2025) considered the online episodic MDP setting, which inherently does not need any notion of data coverage and thus their results are not directly adaptable to our offline setting. Foster et al. (2025) considered an interesting hybrid setting in which the $n$ state-action pairs are still from the offline dataset but $\Omega(n)$ online reward queries and policy switches are allowed; in contrast, in our setting, all reward signals are obtained in a purely offline manner.

**Previous analyses and results in detail.**   Here, we briefly discuss the direct adaptation of previous sample complexity analysis and results (with respect to KL-regularized objectives) to our setting and demonstrate the reason why theirs cannot imply an $\widetilde{O}(\epsilon^{-1})$ sample complexity without all-policy concentrability. In previous analysis of pessimism for unregularized objectives (Jin et al., 2021; Xiong et al., 2022), the sub-optimality gap is decomposed via the performance difference lemma as

follows

$$\begin{aligned}
J(\pi^*) - J(\widehat{\pi}) &= \mathbb{E}_{a \sim \pi^*}[g^*(a)] - \mathbb{E}_{a \sim \widehat{\pi}}[g^*(a)] - \eta^{-1}\mathsf{KL}(\pi^*\|\pi^{\mathsf{ref}}) + \eta^{-1}\mathsf{KL}(\widehat{\pi}\|\pi^{\mathsf{ref}}) \\
&\leq \mathbb{E}_{a \sim \pi^*}[g^*(a)] - \mathbb{E}_{a \sim \widehat{\pi}}[\widehat{g}(a)] - \eta^{-1}\mathsf{KL}(\pi^*\|\pi^{\mathsf{ref}}) + \eta^{-1}\mathsf{KL}(\widehat{\pi}\|\pi^{\mathsf{ref}}) \\
&\leq \mathbb{E}_{a \sim \pi^*}[g^*(a)] - \mathbb{E}_{a \sim \pi^*}[\widehat{g}(a)] - \eta^{-1}\mathsf{KL}(\pi^*\|\pi^{\mathsf{ref}}) + \eta^{-1}\mathsf{KL}(\pi^*\|\pi^{\mathsf{ref}}) \\
&= \mathbb{E}_{a \sim \pi^*}[g^*(a) - \widehat{g}(a)],
\end{aligned}$$

where the first inequality holds due to pessimism and last inequality holds due to $\widehat{\pi}$ is optimal for $\widehat{g}$. Notably, the KL-regularization term is canceled out in the analysis, leading to a loose sample complexity $\widetilde{O}(\epsilon^{-2})$ since the curvature of KL-divergence is not exploited. Specifically, under linear function approximation, this performance gap, obtained by Xiong et al. (2024) becomes

$$J(\pi^*) - J(\pi) \leq \left\| \mathbb{E}_{\rho \times \pi^*}[\phi(s,a)] - \boldsymbol{\nu} \right\|_{\boldsymbol{\Sigma}_{\mathsf{off}}^{-1}} =: \mathsf{RHS},$$

where $\boldsymbol{\nu}$ is the reference vector, $\phi(s,a) \in \mathbb{R}^d$ is the feature map, and $\boldsymbol{\Sigma}_{\mathsf{off}} = \sum_{i=1}^n \phi(s_i, a_i)\phi(s_i, a_i)^\top$ is the sample covariance matrix. However, we can show that RHS can be bounded from *below* by

$$\begin{aligned}
\left\| \mathbb{E}_{(s,a) \sim \rho \times \pi^*}[\phi(s,a)] - \boldsymbol{\nu} \right\| \sqrt{\lambda_{\min}(\boldsymbol{\Sigma}_{\mathsf{off}}^{-1})} &= \left\| \mathbb{E}_{(s,a) \sim \rho \times \pi^*}[\phi(s,a)] - \boldsymbol{\nu} \right\| \lambda_{\max}(\boldsymbol{\Sigma}_{\mathsf{off}})^{-1/2} \\
&\geq \left\| \mathbb{E}_{(s,a) \sim \rho \times \pi^*}[\phi(s,a)] - \boldsymbol{\nu} \right\| \mathrm{tr}(\boldsymbol{\Sigma}_{\mathsf{off}})^{-1/2} \\
&= \left\| \mathbb{E}_{(s,a) \sim \rho \times \pi^*}[\phi(s,a)] - \boldsymbol{\nu} \right\| \left( \sum_{i=1}^n \|\phi(s_i, a_i)\|_2^2 \right)^{-1/2} \\
&= \Omega(n^{-1/2}),
\end{aligned}$$

where $\lambda_{\min}$ and $\lambda_{\max}$ is the minimum and maximum eigenvalue of a matrix, the first inequality holds due to the fact that $\mathbf{x}^\top \boldsymbol{\Sigma} \mathbf{x} \geq \|\mathbf{x}\|_2^2 \lambda_{\min}(\boldsymbol{\Sigma})$ and the second inequality holds due to $\lambda_{\max}(\boldsymbol{\Sigma}) \leq \mathrm{tr}(\boldsymbol{\Sigma})$. Zhao et al. (2024) proposed a two-stage learning algorithm and obtained an $\widetilde{O}(\epsilon^{-1})$ sample complexity for online KL-regularized bandits. The algorithm can be adopted to offline learning by removing the second stage[5] and treat the samples from first stage as the offline dataset. An analogous analysis gives a sample complexity of $\widetilde{O}(D^2\epsilon^{-1})$, where $D^2$ is the all-policy concentrability.

## B  ADDITIONAL DISCUSSION OF RELATION BETWEEN COVERAGE MEASURES

In this section, we provide more illustrations on the relation between two coverage measures, $D_{\pi^*}^2$ and $C^{\pi^*}$. In particular, we provide two cases under linear function approximation, on one of which $D_{\pi^*}^2 = \Theta(dC^{\pi^*})$ and on the other we have $D_{\pi^*}^2 \ll C^{\pi^*}$, where $d$ is the dimension of the function class. We summarized them as two propositions.

**Proposition B.1.** *There exist a KL-regularized linear bandit instance, such that $D_{\pi^*}^2 = \Theta(dC^{\pi^*})$.*

*Proof.* We construct the instance as follows. Let $d = 2A + 1$ be some odd number and consider an $2A + 1$-armed bandit, such that the feature vector of the $i$-th arm, $\phi(a_i) = \mathbf{e}_i \in \mathbb{R}^d$, which has 1 on its $i$-th entry and 0 on all other entries. The reference policy $\pi^{\mathsf{ref}}(a_i) = (2AC)^{-1}$ for $i \in [2A]$ and $\pi^{\mathsf{ref}}(a_{2A+1}) = (C - 1)/C$, where $2C - 1 = e^\eta$. The ground truth reward function $\boldsymbol{\theta}^* = \sum_{i \leq A} \mathbf{e}_i$ and the function class is given by all $\|\boldsymbol{\theta}\|_\infty \leq 1$. By construction, we know that $\pi^*(a_i) \geq \pi^{\mathsf{ref}}(a_i)$ if and only if $i \in [A]$ and its closed form is given by

$$\pi^*(a_i) = \frac{1}{A} \frac{e^\eta}{e^\eta + 2C - 1} = \frac{1}{2A},$$

which gives $C^{\pi^*} = C$. Now we compute the $D_{\pi^*}^2$ of this instance. For all $i \in [A]$, we know that

$$D^2(a_i) = \sup_{\|\boldsymbol{\theta}\|_\infty \leq 2} \frac{\langle \boldsymbol{\theta}, \mathbf{e}_i \rangle^2}{\mathbb{E}_{\pi^{\mathsf{ref}}} \langle \boldsymbol{\theta}, \mathbf{e}_j \rangle^2} = 2CA = \Theta(Cd),$$

---

[5]This can be done by setting the $n$ in their paper to 0.

where the second equation holds with $\boldsymbol{\theta} = \mathbf{e}_i$. Taking expectation over $\pi^*$, we have

$$D_{\pi^*}^2 \geq \sum_{i \in [A]} D^2(a_i) = \Theta(C^{\pi^*} d),$$

which concludes the proof. □

The following proposition provides another instance on which $D_{\pi^*}^2 \ll C^{\pi^*}$.

**Proposition B.2.** For any $C \geq 2$, there exists a KL-regularized linear bandit instance, such that $C^{\pi^*} = C/2$ and $D_{\pi^*}^2 = \Theta(1)$.

*Proof.* We consider the function class of $\boldsymbol{\theta} \in \mathbb{R}^2$ and $\|\boldsymbol{\theta}\| \leq \sqrt{2}$. The instance consists of three arms, where $\phi(a_1) = (1, 0)$, $\phi(a_2) = (0, 1)$, and $\phi(a_3) = (1, 1)$. The ground truth parameter $\boldsymbol{\theta}^* = (1, 1)$. The reference policy is given by $\pi^{\mathsf{ref}}(a_1) = \pi^{\mathsf{ref}}(a_2) = 1/2 - 1/2C$ and $\pi^{\mathsf{ref}}(a_1) = 1/C$, where $C - 1 = e^\eta$. A direct computation yields that

$$\pi^*(a_3) = \frac{e^\eta}{e^\eta + C - 1}, \quad \Rightarrow \quad C^{\pi^*} = C\frac{e^\eta}{e^\eta + C - 1} = \frac{C}{2}.$$

On the other hand, we know that for $i = 1, 2$, we have $D^2(a_i) \leq \pi^{\mathsf{ref}}(a_i)^{-1} \leq 4$. As for $a_3$, since we have $\langle \boldsymbol{\theta}, \phi(a_3) \rangle^2 = \langle \boldsymbol{\theta}, \phi(a_1) + \phi(a_2) \rangle^2 \leq 2 \langle \boldsymbol{\theta}, \phi(a_1) \rangle^2 + 2 \langle \boldsymbol{\theta}, \phi(a_2) \rangle$, which gives that $D^2(a_3) \leq 2D^2(a_1) + 2D^2(a_2) \leq 16$. Therefore, taking expectation over $\pi^*$, we know that $D_{\pi^*}^2 \leq 12$ which is a constant. □

## C EXPERIMENTAL DETAILS

### C.1 LINEAR BANDITS

The linear bandit instance used for Figure 2 has $d = 20$ and $|\mathcal{A}| = 100$. For each arm $a \in \mathcal{A}$, we randomly generate its feature vector $\phi(a) \in \mathbb{R}^d$ such that $\|\phi(a)\| = 1$. We then randomly sample the model parameter $\boldsymbol{\theta}^* \in \mathbb{R}^d$ such that $\|\boldsymbol{\theta}^*\| = 1$ and the expected reward is obtained via $r(a) = \langle \boldsymbol{\theta}^*, \phi(a) \rangle$.

### C.2 REAL-WOLD EXPERIMENTS

MNIST consists of 60000 figures, each of which is of $28 \times 28$ pixels and consists of a handwritten digit in $\{0, \cdots, 9\}$. Here, we consider each image as a context and $\mathcal{A} = \{0, \cdots, 9\}$ for each context. To obtain the feature $\phi(s, a)$, we first use the hidden representation of a classifier to embed each image as a vector in $\mathbb{R}^{10}$. We then follow the approach in Zhou et al. (2020) to obtain the feature of each context-action pair by having $\phi(s, a) = \mathbf{x} \otimes \mathbf{e}_{a+1} \in \mathbb{R}^{100}$, where $\mathbf{x}$ is the output of image encoder and $\otimes$ stands for tensor product.

## D MISSING PROOFS FROM SECTION 2

### D.1 PROOF OF LEMMA 2.9

We first provide the following lemmas of concentration.

**Lemma D.1** (Zhao et al. 2024, Lemma C.1). *For any policy $\pi$ and state-action pairs $\{(s_i, a_i)\}_{i=1}^m$ generated i.i.d. from $\rho \times \pi$, and $\epsilon_c < 1$, with probability at least $1 - \delta$, for any $g_1$ and $g_2$ we have*

$$\mathbb{E}_{\rho \times \pi}\left[\left(g_1(s, a) - g_2(s, a)\right)^2\right] \leq \frac{2}{n} \sum_{i=1}^n \left(g_1(s_i, a_i) - g_2(s_i, a_i)\right)^2 + \frac{32}{3n} \log(2\mathcal{N}_{\mathcal{G}}(\epsilon_c)/\delta) + 10\epsilon_c,$$

*where $\mathcal{N}_{\mathcal{G}}(\epsilon_c)$ is the $\epsilon_c$-covering number of $\mathcal{G}$.*

**Lemma D.2** (Zhao et al. 2024, Lemma C.2). *For arbitrary policy $\pi$ and dataset $\{(s_i, a_i, r_i)\}_{i=1}^m$ generated i.i.d., from the product of $\pi$, $\rho$ and the Bradley-Terry Model; let $\bar{g}$ be the least square estimator of $g^*$, then for any $0 < \epsilon_c < 1$ and $\delta > 0$, with probability at least $1 - \delta$ we have*

$$\sum_{i=1}^n \left(\bar{g}(s_i, a_i) - g^*(s_i, a_i)\right)^2 \leq 16 \log(a\mathcal{N}_{\mathcal{G}}(\epsilon_c)/\delta) + 4n\epsilon_c.$$

Now we are ready to prove Lemma 2.9.

*Proof of Lemma 2.9.* We have the following inequality

$$
\big(\bar{g}(s,a) - g^*(s,a)\big)^2 = \frac{\big(\bar{g}(s,a) - g^*(s,a)\big)^2}{\mathbb{E}_{\pi^{\mathsf{ref}}}\big[\big(\bar{g}(s,a) - g^*(s,a)\big)^2\big]} \mathbb{E}_{\pi^{\mathsf{ref}}}\big[\big(\bar{g}(s,a) - g^*(s,a)\big)^2\big]
$$

$$
\leq \sup_{g_1,g_2 \in \mathcal{G}} \frac{\big(g_1(s,a) - g_2(s,a)\big)^2}{\mathbb{E}_{\pi^{\mathsf{ref}}}\big[\big(g_1(s,a) - g_2(s,a)\big)^2\big]} \mathbb{E}_{\pi^{\mathsf{ref}}}\big[\big(\bar{g}(s,a) - g^*(s,a)\big)^2\big]
$$

$$
= D_{\mathcal{G}}^2((s,a), \pi^{\mathsf{ref}}) \mathbb{E}_{\pi^{\mathsf{ref}}}\big[\big(\bar{g}(s,a) - g^*(s,a)\big)^2\big], \tag{D.1}
$$

where the inequality holds by taking supremum to $g_1, g_2 \in \mathcal{G}$. Now we have

$$
\mathbb{E}_{\pi^{\mathsf{ref}}}\big[\big(\bar{g}(s,a) - g^*(s,a)\big)^2\big] \leq \frac{2}{n} \sum_{i=1}^{n} \big(\bar{g}(s_i,a_i) - g^*(s_i,a_i)\big)^2 + \frac{32}{3n}\log(2\mathcal{N}_{\mathcal{G}}(\epsilon_c)/\delta) + 10\epsilon_c
$$

$$
\leq \frac{2}{n}\big[16\log(\mathcal{N}_{\mathcal{G}}(\epsilon_c)/\delta) + 4n\epsilon_c\big] + \frac{32}{3n}\log(2\mathcal{N}_{\mathcal{G}}(\epsilon_c)/\delta) + 10\epsilon_c
$$

$$
= \frac{128}{3n}\log(2\mathcal{N}_{\mathcal{G}}(\epsilon_c)/\delta) + 18\epsilon_c, \tag{D.2}
$$

where the first inequality holds due to Lemma D.1 and second holds due to Lemma D.2. Plugging (D.2) into (D.1) and setting $\epsilon_c = O(n^{-1})$ complete the proof. $\square$

### D.2 PROOF OF LEMMA 2.14

This proof is extracted from the proof of Zhao et al. (2024, Theorem 3.3) and we present it here for completeness. By definition of our objective in (2.1), we have

$$
J(\pi^*) - J(\pi_g)
$$

$$
= \mathbb{E}_{(s,a)\sim\rho\times\pi^*}\left[g^*(s,a) - \eta^{-1}\log\frac{\pi^*(a|s)}{\pi^{\mathsf{ref}}(a|s)}\right] - \mathbb{E}_{(s,a)\sim\rho\times\pi_g}\left[g^*(s,a) - \frac{1}{\eta}\log\frac{\pi_g(a|s)}{\pi^{\mathsf{ref}}(a|s)}\right]
$$

$$
= \frac{1}{\eta}\mathbb{E}_{(s,a)\sim\rho\times\pi^*}\left[\log\frac{\pi^{\mathsf{ref}}(a|s)\cdot\exp\big(\eta g^*(s,a)\big)}{\pi^*(a|s)}\right] - \frac{1}{\eta}\mathbb{E}_{(s,a)\sim\rho\times\pi_g}\left[\log\frac{\pi^{\mathsf{ref}}(a|s)\cdot\exp\big(\eta g^*(s,a)\big)}{\pi_g(a|s)}\right]
$$

$$
= \frac{1}{\eta}\mathbb{E}_{s\sim\rho}\big[\log Z_{g^*}(s)\big] - \frac{1}{\eta}\mathbb{E}_{s\sim\rho}\big[\log Z_g(s)\big] - \mathbb{E}_{s\sim\rho}\left[\sum_{a\in\mathcal{A}}\pi_g(a|s)\cdot\big(g^*(s,a) - f(s,a)\big)\right],
$$

where for all $g \in \mathcal{G}$ we define $Z_g(\cdot)$ as follows,

$$
Z_g(\cdot) := \sum_{a\in\mathcal{A}}\pi^{\mathsf{ref}}(a|\cdot)\exp\big(\eta g(\cdot,a)\big).
$$

We further denote $\Delta(s,a) = g(s,a) - g^*(s,a)$ and $H_s(g) = \log Z_g(s) - \eta\sum_{a\in\mathcal{A}}\pi_g(a|s)\cdot\Delta(s,a)$. It worth noticing that $\eta^{-1}\mathbb{E}_{s\sim\rho}[H_s(g^*) - H_s(g)] = J(\pi^*) - J(\pi_g)$. Now we take derivative of $H$ with respect to $\Delta(s,a)$,

$$
\frac{\partial H_s(g)}{\partial \Delta(s,a)} = \frac{\partial}{\partial \Delta(s,a)}\left[\log Z_g(s) - \eta\sum_{a\in\mathcal{A}}\pi_g(a|s)\cdot\Delta(s,a)\right]
$$

$$
= \frac{1}{Z_g(s)}\cdot\pi^{\mathsf{ref}}(a|s)\exp\big(\eta\cdot g(s,a)\big)\cdot\eta - \eta\cdot\pi_g(a|s)
$$

$$
- \eta^2\cdot\Delta(s,a)\cdot\frac{\pi^{\mathsf{ref}}(a|s)\cdot\exp\big(\eta\cdot g(s,a)\big)}{Z_g(s)} + \eta^2\cdot\Delta(s,a)\cdot\frac{\big[\pi^{\mathsf{ref}}(a|s)\cdot\exp\big(\eta\cdot g(s,a)\big)\big]^2}{[Z_g(s)]^2}
$$

$$
+ \eta\sum_{a'\in\mathcal{A}\setminus\{a\}}\frac{\pi^{\mathsf{ref}}(a'|x)\cdot\exp\big(\eta\cdot g(s,a')\big)}{Z_g(s)}\cdot\eta\cdot\Delta(s,a')\cdot\frac{\pi^{\mathsf{ref}}(a|s)\cdot\exp\big(\eta\cdot g(s,a)\big)}{Z_g(s)}
$$

$$
= -\eta^2\pi_g(a|s)\Delta(s,a) + \eta^2[\pi_g(a|s)]^2\cdot\Delta(s,a) + \eta^2\sum_{a'\in\mathcal{A}\setminus\{a\}}\pi_g(a'|x)\pi_g(a|s)\Delta(s,a').
$$

Therefore, by mean value theorem, there exists $\gamma \in [0, 1]$ and $g_\gamma = \gamma g + (1 - \gamma)g^*$ such that

$$H_s(g) - H_s(g^*) = -\eta^2 \gamma \sum_{a \in \mathcal{A}} \pi_{g_\gamma}(a|s)\Delta(s, a)^2 + \gamma\eta^2 \sum_{a_1 \in \mathcal{A}} \sum_{a_2 \in \mathcal{A}} \pi_{g_\gamma}(a_1|x)\pi_{g_\gamma}(a_2|x)\Delta(s, a_1)\Delta(s, a_2)$$

$$= -\eta^2 \gamma \mathbb{E}_{a \sim \pi_{g_\gamma}}\left[\left(g^*(s, a) - g(s, a)\right)^2\right] + \gamma\eta^2\left(\mathbb{E}_{a \sim \pi_{g_\gamma}}\left[\left(g^*(s, a) - g(s, a)\right)\right]\right)^2$$

$$\geq -\eta^2 \mathbb{E}_{a \sim \pi_{g_\gamma}}\left[\left(g^*(s, a) - g(s, a)\right)^2\right],$$

where the inequality holds by omitting the second term and $\gamma \leq 1$. Now taking expectation over $\rho$, we have

$$J(\pi^*) - J(\pi_g) = \eta^{-1}\mathbb{E}_{s \sim \rho}[H_s(g^*) - H_s(g)]$$

$$\leq \eta\mathbb{E}_{(s,a) \sim \rho_{g_\gamma}}\left[\left(g^*(s, a) - g(s, a)\right)^2\right],$$

which concludes the proof.

### D.3  PROOF OF LEMMA 2.15

*Proof of Lemma 2.15.* We define $Y = -X$. Then it suffices to show that the covariance between $Y$ and $Y^2$ is

$$\mathrm{Cov}(Y, Y^2) = \mathbb{E}[Y^3] - \mathbb{E}[Y^2]\mathbb{E}[Y]$$

$$\geq \left(\mathbb{E}[Y^2]\right)^{3/2} - \mathbb{E}[Y^2]\mathbb{E}[Y]$$

$$= \left(\mathbb{E}[Y^2]\right)\left(\sqrt{\mathbb{E}[Y^2]} - \mathbb{E}[Y]\right)$$

$$\geq 0,$$

where both inequalities follow from Jensen's inequality. $\qquad\square$

### D.4  PROOF OF THEOREM 2.10

To start with, we first define the following quantities. For all $\gamma \in [0, 1]$, we define $g_\gamma := \gamma\widehat{g} + (1 - \gamma)g^*$ and denote

$$\pi_\gamma(\cdot|s) \propto \pi^{\mathsf{ref}}(\cdot|s)\exp\left(\eta g_\gamma(s, \cdot)\right), \forall s \in \mathcal{S};$$

$$G(\gamma) := \mathbb{E}_{\rho \times \pi_\gamma}\left[\left(\widehat{g} - g^*\right)^2(s, a)\right].$$

The key to our analysis is the monotonicity of the function $G(\gamma)$ in $\gamma$, which is formally stated in the following lemma.

**Lemma D.3.** On event $\mathcal{E}$, $0 \in \mathrm{argmax}_{\gamma \in [0,1]} G(\gamma)$.

*Proof.* For simplicity, we use $\triangle(s, a)$ to denote $\left(\widehat{g} - g^*\right)(s, a)$ in *this* proof. Then we know that $\triangle(s, a) \leq 0$ for all $(s, a) \in \mathcal{S} \times \mathcal{A}$ on event $\mathcal{E}$. The most direct way to prove is to take derivative of $G$ with respect to $\gamma$, which corresponds to the policy gradient (Sutton et al., 1999) of $\pi_\gamma$ and thus implying a favorable structure. A direct calculation yields that

$$= \mathbb{E}_{\rho \times \pi_\gamma}\left[\nabla_\gamma \log \pi_\gamma(a|s)\triangle(s, a)^2\right]$$

$$= \eta\mathbb{E}_\rho\mathbb{E}_{a \sim \pi_\gamma}\left[\triangle^2(s, a)\left(\triangle(s, a) - \mathbb{E}_{a' \sim \pi_\gamma}[\triangle(s, a')]\right)\right]$$

$$= \eta\mathbb{E}_\rho\left[\mathbb{E}_{\pi_\gamma}\left[\triangle^3(s, a)\right] - \mathbb{E}_{\pi_\gamma}\left[\triangle^2(s, a)\right]\mathbb{E}_{\pi_\gamma}\left[\triangle(s, a)\right]\right]$$

$$\leq 0,$$

where $\mathbb{E}_\rho$ is the shorthand of $\mathbb{E}_{s \sim \rho}$, $\mathbb{E}_{\pi_\gamma}$ is the shorthand of $\mathbb{E}_{a \sim \pi_\gamma}$, the first equation is derived from standard policy gradient and the inequality holds conditioned on the event $\mathcal{E}(\delta)$ due to Lemma 2.15 and Lemma 2.15. $\qquad\square$

Now we are ready to prove Theorem 2.10.

*Proof of Theorem 2.10.* Following the proof of Zhao et al. (2024, Theorem 3.3), we know that there exists $\bar{\gamma} \in [0, 1]$ such that

$$J(\pi^*) - J(\widehat{\pi}) \leq \eta G(\bar{\gamma}) \leq \eta G(0), \tag{D.3}$$

where the first inequality holds due to Lemma 2.14 and the second inequality holds due to the event $\mathcal{E}$ and Lemma D.3. The term $G(0)$ can be further bounded with the $D^2$-based concentrability as follows

$$\begin{aligned}
G(0) &= \eta \mathbb{E}_{(s,a) \sim \rho \times \pi^*} \left[ \left( \widehat{g} - g^* \right)^2 (s, a) \right] \\
&\leq 4\eta \mathbb{E}_{(s,a) \sim \rho \times \pi^*} [\Gamma_n^2(s, a)] \\
&= 4\eta \beta^2 \mathbb{E}_{(s,a) \sim \rho \times \pi^*} \left[ D_{\mathcal{F}}^2((s, a); \pi^{\mathsf{ref}}) \right] \\
&= \widetilde{O}(\eta D_{\pi^*}^2 n^{-1} \log_{\mathcal{G}}(\epsilon_c)), \tag{D.4}
\end{aligned}$$

where the second inequality holds conditioned on $\mathcal{E}(\delta)$ because of Lemma D.3, and the last inequality follows from the definition of $\mathcal{E}(\delta)$ together with Line 2. By Lemma 2.9, we know that event $\mathcal{E}$ holds with probability at least $1 - \delta$, which finishes the proof. $\qquad\square$

## D.5 Proof of Theorem 2.11

*Proof of Theorem 2.11.* We consider the family of contextual bandits with $S := |\mathcal{S}|, A := |\mathcal{A}| < \infty$ and reward function in some function class $\mathcal{G}$ composed of function $\mathcal{S} \times \mathcal{A} \to [0, 1]$ as follows.

$$\mathrm{CB}_{\mathcal{G}} := \{(\mathcal{S}, \mathcal{A}, \rho, r, \pi^{\mathsf{ref}}, \eta) : r \in \mathcal{G}, \rho \in \Delta(\mathcal{S}), \pi^{\mathsf{ref}} \in \Delta(\mathcal{A}|\mathcal{S})\}. \tag{D.5}$$

Our goal is to prove the following statement. Fixing any $S \geq 1, \eta > 4 \log 2$ and $C^* \in (2, \exp(\eta/4)]$, then for any estimator $\mathcal{D} \mapsto \widehat{\pi} \in \Delta(\mathcal{A}|\mathcal{S})$, for any $n \geq 16 S C^*$, there exist some function class $\mathcal{G}$, such that $\exists \, \mathsf{inst} = (\mathcal{S}, \mathcal{A}, \rho, r, \pi^{\mathsf{ref}}, \eta) \in \mathrm{CB}_{\mathcal{G}}$ with single-policy concentrability $C^{\pi^*} \leq C^*$, regularization coefficient $\eta$, $|\mathcal{S}| = S = \Theta(\log |\mathcal{G}|)$, and

$$\mathrm{SubOpt}_{\mathrm{RKL}}(\widehat{\pi}; \mathsf{inst}) \gtrsim \min\{\eta S C^* n^{-1}, (S C^*)^{1/2} n^{-1/2}\}. \tag{D.6}$$

Since $\log |\mathcal{G}| \geq \log \mathcal{N}_{\mathcal{G}}(\epsilon)$ for any $\epsilon \in (0, 1)$, equation (D.6) yields the desired bound.

We set $\mathcal{S} = [S], \mathcal{A} = \{\pm 1\}, \rho = \mathsf{Unif}(\mathcal{S})$, and the reference policy to be

$$\forall s \in \mathcal{S}, \pi^{\mathsf{ref}}(-1|s) = C^{-1}, \pi^{\mathsf{ref}}(+1|s) = 1 - C^{-1};$$

where $C \geq 1$ is a parameter to be specified later. We construct $2^S$ Bernoulli reward functions, in particular, $\forall \tau \in \{\pm 1\}^S$, the mean function $r_\tau$ of the reward (indexed by $\tau$) is defined as

$$r_\tau(s, -1) = 0.5 + \tau_s \delta, r_\tau(s, +1) = 0.5 - \alpha$$

for any state $s \in \mathcal{S}$, where $\alpha \in (0, 1/2)$ and $\delta \in (0, 1/4]$ will be specified later. We omit the RKL subscript in the following argument when it is clear in context. By (2.2), the optimal policy $\pi_\tau^*$ under $r_\tau$ is

$$\forall s \in \mathcal{S}, \pi_\tau^*(-1|s) = \frac{\exp\left(\eta(\alpha + \tau_s \delta)\right)}{\exp\left(\eta(\alpha + \tau_s \delta)\right) + C - 1}, \pi_\tau^*(+1|s) = \frac{C - 1}{\exp\left(\eta(\alpha + \tau_s \delta)\right) + C - 1}. \tag{D.7}$$

Since $C^* \leq \exp(\eta/4)$, we assign $C = C^*$ and $\alpha = \eta^{-1} \log(C - 1) \Leftrightarrow C - 1 = \exp(\eta \alpha)$, which gives

$$\begin{aligned}
\forall s \in \mathcal{S}, \frac{\pi_\tau^*(-1|s)}{\pi^{\mathsf{ref}}(-1|s)} &\leq C \frac{\exp(\eta(\alpha + \tau_s \delta))}{C - 1 + \exp(\eta(\alpha + \tau_s \delta))} = C \frac{\exp(\eta \tau_s \delta)}{1 + \exp(\eta \tau_s \delta)} \leq C = C^*; \\
\forall s \in \mathcal{S}, \frac{\pi_\tau^*(+1|s)}{\pi^{\mathsf{ref}}(+1|s)} &= \frac{C}{C - 1} \cdot \frac{1}{\exp(\eta \tau_s \delta) + 1} \leq C = C^*;
\end{aligned}$$

where the last inequality is due to the assumption $C^* \geq 2$. Therefore, we obtain

$$\max_{\tau \in \{\pm 1\}^S} C^{\pi_\tau^*} \leq C^*. \tag{D.8}$$

We will abuse the notation $\mathrm{SubOpt}(\widehat{\pi}; \tau) := \mathrm{SubOpt}(\widehat{\pi}; r_\tau)$. Since $\rho = \mathsf{Unif}(\mathcal{S})$,

$$\mathrm{SubOpt}(\widehat{\pi}; \tau) = \frac{1}{S} \sum_{s=1}^{S} \mathrm{SubOpt}_s(\widehat{\pi}; \tau), \tag{D.9}$$

where

$$
\begin{aligned}
\mathrm{SubOpt}_s(\widehat{\pi}; \tau) &= \langle \pi_\tau^*(\cdot|s), r_\tau(s, \cdot) - \eta^{-1} \log \frac{\pi_\tau^*(\cdot|s)}{\pi^{\mathsf{ref}}(\cdot|s)} \rangle - \langle \widehat{\pi}(\cdot|s), r_\tau(s, \cdot) - \eta^{-1} \log \frac{\widehat{\pi}(\cdot|s)}{\pi^{\mathsf{ref}}(\cdot|s)} \rangle \\
&= \frac{1}{\eta} \mathbb{E}_{a \sim \pi_\tau^*(\cdot|s)} \left[ \log \frac{\pi^{\mathsf{ref}}(a|s) \cdot \exp\left(\eta r_\tau(s, a)\right)}{\pi_\tau^*(a|s)} \right] \\
&\quad - \frac{1}{\eta} \mathbb{E}_{a \sim \widehat{\pi}(\cdot|s)} \left[ \log \frac{\pi^{\mathsf{ref}}(a|s) \cdot \exp\left(\eta r_\tau(s, a)\right)}{\widehat{\pi}(a|s)} \right] \\
&= \frac{1}{\eta} \mathbb{E}_{a \sim \pi_\tau^*(\cdot|s)} \left[ \log \left( \sum_{b \in \mathcal{A}} \pi^{\mathsf{ref}}(b|s) \cdot \exp\left(\eta r_\tau(s, b)\right) \right) \right] \\
&\quad - \frac{1}{\eta} \mathbb{E}_{a \sim \widehat{\pi}(\cdot|s)} \left[ \log \frac{\pi^{\mathsf{ref}}(a|s) \cdot \exp\left(\eta r_\tau(s, a)\right)}{\widehat{\pi}(a|s)} \right] \\
&= \frac{1}{\eta} \mathbb{E}_{a \sim \widehat{\pi}(\cdot|s)} \left[ \log \frac{\pi^{\mathsf{ref}}(a|s) \cdot \exp\left(\eta r_\tau(s, a)\right)}{\pi_\tau^*(a|s)} - \log \frac{\pi^{\mathsf{ref}}(a|s) \cdot \exp\left(\eta r_\tau(s, a)\right)}{\widehat{\pi}(a|s)} \right] \\
&= \eta^{-1} \mathsf{KL}\left(\widehat{\pi} \| \pi_\tau^*\right). \tag{D.10}
\end{aligned}
$$

We write $\tau \sim_s \tau'$ if $\tau, \tau' \in \{\pm 1\}^{\mathcal{S}}$ differ in only the $s$-th coordinate and $\tau \sim \tau'$ if $\exists s \in \mathcal{S}, \tau \sim_s \tau'$. By (D.10), $\forall s \in \mathcal{S}, \forall \tau, \tau' \in \{\pm 1\}^{\mathcal{S}}$ with $\tau \sim_s \tau'$,

$$
\begin{aligned}
& \mathrm{SubOpt}_s(\widehat{\pi}; \tau) + \mathrm{SubOpt}_s(\widehat{\pi}; \tau') \\
&= \eta^{-1} \mathsf{KL}\left(\widehat{\pi} \| \pi_\tau^*\right) + \eta^{-1} \mathsf{KL}\left(\widehat{\pi} \| \pi_{\tau'}^*\right) \\
&= 2\eta^{-1} \sum_{a \in \mathcal{A}} \widehat{\pi}(a|s) \log \frac{\widehat{\pi}(a|s)}{\sqrt{\pi_\tau^*(a|s)\pi_{\tau'}^*(a|s)}} \\
&= 2\eta^{-1} \mathsf{KL}\left(\widehat{\pi}(\cdot|s) \| \bar{\pi}_{\tau,\tau'}(\cdot|s)\right) - 2\eta^{-1} \mathbb{E}_{a \sim \widehat{\pi}(\cdot|s)} \log \left( \sum_{b \in \mathcal{A}} \sqrt{\pi_\tau^*(b|s)\pi_{\tau'}^*(b|s)} \right) \\
&\geq -2\eta^{-1} \log \left( \sum_{b \in \mathcal{A}} \sqrt{\pi_\tau^*(b|s)\pi_{\tau'}^*(b|s)} \right) \\
&= \frac{1}{\eta} \log \frac{(\exp(\eta\delta) + 1)(\exp(-\eta\delta) + 1)}{4}, \tag{D.11}
\end{aligned}
$$

where $\bar{\pi}(\cdot|s) = \sqrt{\pi_\tau^*(\cdot|s)\pi_{\tau'}^*(\cdot|s)} / \sum_{b \in \mathcal{A}} \sqrt{\pi_\tau^*(b|s)\pi_{\tau'}^*(b|s)}$ for every $s \in \mathcal{S}$, the inequality is due to the non-negativity of KL divergence, and the last equality follows from (D.7) together with the design choice $C - 1 = \exp(\eta\alpha)$.

**Case $\eta\delta \leq 2$.** Recall that $\forall x \in \mathbb{R}, (e^x + e^{-x})/2 - 1 = x^2 \sum_{k=0}^{\infty} \frac{x^{2k}}{(2k+2)!} \geq x^2/2$, which implies

$$(\text{D.11}) = \frac{1}{\eta} \log \left( 1 + \frac{1}{2}\left( \frac{e^{\eta\delta} + e^{-\eta\delta}}{2} - 1 \right) \right) \geq \frac{1}{\eta} \log \left( 1 + \frac{\eta^2\delta^2}{4} \right) \geq \frac{1}{\eta} \cdot \frac{\eta^2\delta^2/4}{2} = \eta\delta^2/8. \tag{D.12}$$

Here, the last inequality is due to $\eta^2\delta^2/4 \leq 1$ and $\forall x \in [0, 1], \log(1 + x) \geq x/2$.

**Case $\eta\delta > 2$.** We have $-\eta^{-1}2 \log 2 \geq -\delta \log 2$, which implies the following bound.

$$(\text{D.11}) \geq \frac{1}{\eta} \log \frac{\exp(\eta\delta) + 1}{4} \geq \frac{\eta\delta - 2\log 2}{\eta} = \delta - \eta^{-1}2\log 2 \geq (1 - \log 2)\delta \geq 3\delta/10. \tag{D.13}$$

In summary, (D.12) and (D.13) imply that $\forall s \in \mathcal{S}, \forall \tau, \tau' \in \{\pm 1\}^{\mathcal{S}}$ with $\tau \sim_s \tau'$,

$$\mathrm{SubOpt}_s(\widehat{\pi}; \tau) + \mathrm{SubOpt}_s(\widehat{\pi}; \tau') \geq \frac{\eta\delta^2}{8} \wedge \frac{3\delta}{10}. \tag{D.14}$$

Let $P_\tau$ be the distribution of $(s, a, y)$ where $s \sim \rho, a \sim \pi^{\text{ref}}(\cdot|s)$, and $y \sim \text{Bern}(r_\tau(s,a))$. Then $\forall x \in \mathcal{S} \; \forall \tau, \tau' \in \{\pm 1\}^\mathcal{S}$ with $\tau \sim_x \tau'$,

$$
\begin{aligned}
\text{KL}\left(P_\tau \| P_{\tau'}\right) &= \frac{1}{S} \sum_{s,a} \pi^{\text{ref}}(a|s) \text{KL}\left(\text{Bern}(r_\tau(s,a)) \| \text{Bern}(r_{\tau'}(s,a))\right) \\
&= \frac{1}{S} \cdot C^{-1} \text{KL}\left(\text{Bern}(r_\tau(x,-1)) \| \text{Bern}(r_{\tau'}(x,-1))\right) \\
&\leq \frac{4\delta^2}{SC(0.25 - \delta^2)} \leq \frac{16\delta^2}{3SC},
\end{aligned}
\tag{D.15}
$$

where we use the requirement $\delta \leq 1/4$ and $\text{KL}\left(\text{Bern}(p) \| \text{Bern}(q)\right) \leq (p-q)^2 / (q(1-q))$. Then let $P_{\mathcal{D}_\tau}$ be the distribution of $\mathcal{D}$ given the mean reward function $r_\tau$, we employ (D.15) to get

$$
\text{KL}\left(P_{\mathcal{D}_\tau} \| P_{\mathcal{D}_{\tau'}}\right) = n \text{KL}\left(P_\tau \| P_{\tau'}\right) \leq \frac{16n\delta^2}{3SC}.
\tag{D.16}
$$

Since $n \geq 16SC^* = 16SC$ by design, we can set $\delta = \sqrt{SC/n}$ (which ensures $\delta \leq 1/4$) to obtain

$$
\begin{aligned}
\sup_{\text{inst}} \text{SubOpt}(\widehat{\pi}; \text{inst}) &\geq \sup_{\tau \in \{\pm 1\}^\mathcal{S}} \text{SubOpt}(\widehat{\pi}; \tau) \\
&\geq \frac{1}{S} \cdot S \cdot \frac{1}{4} \cdot \left(\frac{\eta\delta^2}{8} \wedge \frac{3\delta}{10}\right) \min_{\tau \sim \tau'} \exp\left(-\text{KL}\left(P_{\mathcal{D}_\tau} \| P_{\mathcal{D}_{\tau'}}\right)\right) \\
&\geq \left(\frac{\eta SC^*}{32n} \wedge \frac{3\sqrt{SC^*}}{40\sqrt{n}}\right) \exp(-16/3) \gtrsim \frac{\eta SC^*}{n} \wedge \sqrt{\frac{SC^*}{n}}.
\end{aligned}
$$

where the $S^{-1}$ in the second inequality comes from (D.9), the second inequality is by substituting (D.14) into Assouad's Lemma (Lemma H.3), and the last inequality is due to (D.16). $\qquad\square$

# E  MISSING PROOF FROM SECTION 3

## E.1  PROOF OF THEOREM 3.2

Before coming to the proof, we first introduce some useful properties. The following properties characterize the convexity of $f$-divergence when $f$ is (strongly) convex.

The strong-convexity of $f$ implies that the corresponding $f$-divergence, $D_f(\cdot||\pi^{\text{ref}})$ is also strongly convex with respect to all $\pi : \mathcal{S} \to \Delta(\mathcal{A})$ supported by $\pi^{\text{ref}}$.

**Proposition E.1.** Given context $s$, $D_f(\pi(\cdot|s)||\pi^{\text{ref}}(\cdot|s))$ is strict convex with respect to $\pi$ if $f$ is strictly convex.

**Proposition E.2.** Given context $s$, $\pi(\cdot|s) \mapsto D_f(\pi(\cdot|s)||\pi^{\text{ref}}(\cdot|s))$ is $4\alpha$-strong convex with respect to the metric TV if $f$ is $\alpha$-strongly convex.

*Proof of Proposition E.2.* We first show the gradient of $D_f$ with respect to $\pi$.

$$
\frac{\partial D_f(\pi||\pi^{\text{ref}})}{\pi(a)} = \frac{\partial}{\partial \pi(a)} \sum_{b \in \mathcal{A}} \pi^{\text{ref}}(b) f\left(\frac{\pi(b)}{\pi^{\text{ref}}(b)}\right) = f'\left(\frac{\pi(a)}{\pi^{\text{ref}}(a)}\right).
$$

Now consider $\pi_1, \pi_2 \in \Delta(\mathcal{A})$ supported by $\pi^{\mathsf{ref}}$.

$$D_f(\pi_1 || \pi^{\mathsf{ref}}) - D_f(\pi_2 || \pi^{\mathsf{ref}}) - \langle \pi_1 - \pi_2, \nabla D_f(\pi_2 || \pi^{\mathsf{ref}}) \rangle$$

$$= \sum_{a \in \mathcal{A}} \pi^{\mathsf{ref}}(a) \left( f\left( \frac{\pi_1(a)}{\pi^{\mathsf{ref}}(a)} \right) - f\left( \frac{\pi_2(a)}{\pi^{\mathsf{ref}}(a)} \right) \right) - \sum_{a \in \mathcal{A}} (\pi_1(a) - \pi_2(a)) f'\left( \frac{\pi_2(a)}{\pi^{\mathsf{ref}}(a)} \right)$$

$$= \sum_{a \in \mathcal{A}} \pi^{\mathsf{ref}}(a) \left( f\left( \frac{\pi_1(a)}{\pi^{\mathsf{ref}}(a)} \right) - f\left( \frac{\pi_2(a)}{\pi^{\mathsf{ref}}(a)} \right) - \left( \frac{\pi_1(a)}{\pi^{\mathsf{ref}}(a)} - \frac{\pi_2(a)}{\pi^{\mathsf{ref}}(a)} \right) f'\left( \frac{\pi_2(a)}{\pi^{\mathsf{ref}}(a)} \right) \right)$$

$$\geq \frac{\alpha}{2} \sum_{a \in \mathcal{A}} \pi^{\mathsf{ref}}(a) \left( \frac{\pi_1(a)}{\pi^{\mathsf{ref}}(a)} - \frac{\pi_2(a)}{\pi^{\mathsf{ref}}(a)} \right)^2$$

$$= \frac{\alpha}{2} \sum_{a \in \mathcal{A}} \frac{1}{\pi^{\mathsf{ref}}(a)} (\pi_1(a) - \pi_2(a))^2$$

$$\geq \frac{\alpha}{2} \left( \sum_{a \in \mathcal{A}} |\pi_1(a) - \pi_2(a)| \right)^2,$$

where the first inequality holds due to $f$'s strong convexity and the second holds due to Cauchy–Schwarz. The proof finishes since $\|\pi_1 - \pi_2\|_1 = 2\mathsf{TV}(\pi_1 \| \pi_2)$. $\qquad\square$

We first introduce some notation and important properties concerning the convex conjugate of functions. Given some context $s$, we denote the regularization term as $H_s(\pi) = \eta^{-1} D_f(\pi(\cdot|s) \| \pi^{\mathsf{ref}}(\cdot|s))$. We use $H_s^*(r)$ to denote the convex conjugate of $H_s$, which is defined as

$$H_s^*(r) = \sup_{\pi \in \mathcal{S} \to \Delta^{|\mathcal{A}|}} \{ \langle \pi(\cdot|s), r(s, \cdot) \rangle - H_s(\pi) \}.$$

We have the following properties for the convex conjugate. The first property gives the gradient of convex conjugate (see, e.g., Zhou 2018, Lemma 5).

**Proposition E.3.** Given context $s$, and convex $f$, let $\pi_r \in \arg\max_\pi \{ \langle \pi(\cdot|s), r(s, \cdot) \rangle - H_s(\pi) \}$ for some $r$, then the gradient of $H_s^*$ is given by $\nabla H_s^*(r) = \pi_r(\cdot|s)$.

We also need some properties of $\nabla^2 H_s^*$, the Hessian matrix of the convex conjugate function. We first give the Hessian matrix of the original function $H_s$ as follows.

$$\nabla^2 H_s(\pi) = \eta^{-1} \mathrm{diag}\left( \frac{f''\left( \frac{\pi(a_1|s)}{\pi^{\mathsf{ref}}(a_1|s)} \right)}{\pi^{\mathsf{ref}}(a_1|s)}, \cdots, \frac{f''\left( \frac{\pi(a_{|\mathcal{A}|}|s)}{\pi^{\mathsf{ref}}(a_{|\mathcal{A}|}|s)} \right)}{\pi^{\mathsf{ref}}(a_{|\mathcal{A}|}|s)} \right). \tag{E.1}$$

Furthermore, when $f$ is $\alpha$-strongly convex, we have

$$\nabla^2 H_s(\pi) \succeq \alpha \eta^{-1} \mathrm{diag}\left( \pi^{\mathsf{ref}}(a_1|s)^{-1}, \cdots, \pi^{\mathsf{ref}}(a_{|\mathcal{A}|}|s)^{-1} \right).$$

The following lemma, which gives an estimate of $\nabla^2 H_s^*$, is the pivot of the proof.

**Lemma E.4.** For any reward $r : \mathcal{S} \times \mathcal{A} \to [0, 1]$, we have

$$\nabla^2 H_s^*(r) \preceq \alpha^{-1} \eta \mathrm{diag}\left( \pi^{\mathsf{ref}}(a_1|s), \cdots, \pi^{\mathsf{ref}}(a_{|\mathcal{A}|}|s) \right).$$

*Proof of Lemma E.4.* Given reward function $r : \mathcal{S} \times \mathcal{A} \to [0, 1]$, we consider

$$\pi_r \in \arg\max_{\pi \in \mathcal{S} \to \Delta^{|\mathcal{A}|}} \{ \langle \pi(\cdot|s), r(\cdot|s) \rangle - H_s(\pi) \}.$$

From (E.1) we know that $\nabla^2 H_s(\pi_r)$ is invertible. Therefore, by Penot 1994, Proposition 3.2, we have $\nabla^2 H_s^*(r) \preceq (\nabla^2 H_s(\pi_r))^{-1}$. Since $f$ is $\alpha$-strongly convex, we have

$$\nabla^2 H_s^*(r) \preceq \alpha^{-1} \eta \mathrm{diag}\left( \pi^{\mathsf{ref}}(a_1|s), \cdots, \pi^{\mathsf{ref}}(a_{|\mathcal{A}|}|s) \right),$$

which finishes the proof. $\qquad\square$

Now we are ready to prove Theorem 3.2.

*Proof of Theorem 3.2.* Consider our estimation $\bar{g}$ which approximates the ground truth reward function $g^*$, we know that

$$\widehat{\pi} = \underset{\pi \in \mathcal{S} \to \Delta(\mathcal{A})}{\operatorname{argmax}} \left\{ \mathbb{E}_{(s,a) \sim \rho \times \pi}[\bar{g}(s,a)] - \eta^{-1} \mathbb{E}_{s \sim \rho}\left[D_f(\pi \| \pi^{\mathsf{ref}})\right] \right\}.$$

We have the following sub-optimality decomposition

$$
\begin{aligned}
J(\pi^*) - J(\widehat{\pi}) &= \mathbb{E}_{s \sim \rho}\left[ \mathbb{E}_{a \sim \pi^*}[g^*(s,a)] - \mathbb{E}_{a \sim \widehat{\pi}}[g^*(s,a)] - \eta^{-1}\left[D_f(\pi^* \| \pi^{\mathsf{ref}}) - D_f(\widehat{\pi} \| \pi^{\mathsf{ref}})\right] \right] \\
&= \mathbb{E}_{s \sim \rho}\left[ H_s^*(g^*) - H_s^*(\bar{g}) - \langle \widehat{\pi}, g^* - \bar{g} \rangle \right] \\
&= \mathbb{E}_{s \sim \rho}\left[ H_s^*(g^*) - H_s^*(\bar{g}) - \langle \nabla H_s^*(\bar{g}), g^* - \bar{g} \rangle \right] \\
&= \mathbb{E}_{s \sim \rho}[(g^* - \bar{g})^{\top} \nabla^2 H_s^*(\widetilde{g})(g^* - \bar{g})],
\end{aligned}
$$

where $\widetilde{g} = \gamma g^* + (1 - \gamma)\bar{g}$ and $\gamma \in [0,1]$ and the last equation holds due to Taylor's expansion. Now, for any $\delta \in (0,1)$ and $\epsilon_c > 0$, with probability at least $1 - \delta$

$$
\begin{aligned}
J(\pi^*) - J(\widehat{\pi}) &= \mathbb{E}_{s \sim \rho}[(g^* - \bar{g})^{\top} \nabla^2 H_s^*(\widetilde{g})(g^* - \bar{g})] \\
&\leq \alpha^{-1} \eta \mathbb{E}_{s \sim \rho}\left[ (g^* - \bar{g})^{\top} \operatorname{diag}\big(\pi^{\mathsf{ref}}(a_1 | s), \cdots, \pi^{\mathsf{ref}}(a_{|\mathcal{A}|} | s)\big)(g^* - \bar{g}) \right] \\
&= \alpha^{-1} \eta \mathbb{E}_{(s,a) \sim \rho \times \pi^{\mathsf{ref}}}\left[ \big(g^*(s,a) - \bar{g}(s,a)\big)^2 \right] \\
&\leq \alpha^{-1} \eta \left( \frac{128}{3n} \log(2 \mathcal{N}_{\mathcal{G}}(\epsilon_c)/\delta) + 18 \epsilon_c \right),
\end{aligned}
$$

where the first inequality holds due to Lemma E.4 and last inequality holds due to equation (D.2). Setting $\epsilon_c = O(n^{-1})$ completes the proof. □

## E.2 PROOF OF THEOREM 3.4

We first provide the following lemma that gives the close form of optimal policy under $\chi^2$-divergence regularization.

**Lemma E.5** (Huang et al. (2025a, Lemma G.2)). *Let $\pi^*$ be the optimal policy of $\chi^2$-divergence regularized objective with reward function $r$, then $\pi^*$ has the closed form*

$$\pi^*(\cdot) = \pi^{\mathsf{ref}}(\cdot) \max\left\{0, \eta(r(\cdot) - \lambda)\right\}, \text{ where } \sum_{a \in \mathcal{A}} \pi_{f\mathrm{div}}^*(a) = 1.$$

By Proposition E.2, $\pi_{f\mathrm{div}}^* = \operatorname{argmax}_{\pi \in \Delta(\mathcal{A})} J_{f\mathrm{div}}(\pi)$ is unique. The sub-optimality gap for $f$-divergence is consequently defined as

$$\operatorname{SubOpt}_{f\mathrm{div}}(\cdot) := \operatorname{SubOpt}_{f\mathrm{div}}(\cdot; \mathcal{A}, r, \pi^{\mathsf{ref}}) = J_{f\mathrm{div}}(\pi_{f\mathrm{div}}^*) - J_{f\mathrm{div}}(\cdot). \tag{E.2}$$

Now we are ready to prove Theorem 3.4.

*Proof of Theorem 3.4.* We still consider the family of contextual bandits $\mathrm{CB}_{\mathcal{G}}$ given by (D.5). We, still, aim to prove the following statement. Fixing any $S \geq 32 \log 2$, $\eta > 4 \log 2$ and $\alpha$, we set $f(x) := \alpha(x-1)^2/2$, then for any estimator $\mathcal{D} \mapsto \widehat{\pi} \in \Delta(\mathcal{A}|\mathcal{S})$, for any $n$ sufficiently large, there exist some function class $\mathcal{G}$, such that $\exists \, \mathrm{inst} = (\mathcal{S}, \mathcal{A}, \rho, r, \pi^{\mathsf{ref}}, \eta) \in \mathrm{CB}_{\mathcal{G}}$ with $|\mathcal{S}| = S = \Theta(\log |\mathcal{G}|)$, and

$$\operatorname{SubOpt}_{f\mathrm{div}}(\widehat{\pi}; \mathrm{inst}) \gtrsim \alpha^{-1} \eta S n^{-1}. \tag{E.3}$$

Since $\log |\mathcal{G}| \geq \log \mathcal{N}_{\mathcal{G}}(\epsilon)$ for any $\epsilon \in (0,1)$, equation (E.3) yields the desired bound.

We again omit subscripts $f\mathrm{div}$ when it is clear in context. We set $\mathcal{S} = [S]$, $\mathcal{A} = \{-1, +1\}$, and $\rho = \mathsf{Unif}(\mathcal{S})$. For all $s \in \mathcal{S}$, $\pi^{\mathsf{ref}} = \mathsf{Unif}(\mathcal{A})$. We further consider the following reward function class. We leverage Lemma H.4 and obtain a set $\mathcal{V} \in \{-1, +1\}^S$ such that (1) $|\mathcal{V}| \geq \exp(S/8)$ and (2) for any $v, v' \in \mathcal{V}, v \neq v'$, one has $\|v - v'\|_1 \geq S/2$. We construct the following reward

function class where the reward follows Bernoulli distribution and the mean functions are given by the function class

$$\mathcal{G} = \{r_v(s, -1) = 1/2 + v_s\delta, r_{v'}(s, +1) = 1/2 + v'_s\delta, \ \forall s \in \mathcal{S} | v \in \mathcal{V}\},$$

where $\delta \in (0, \eta^{-1}\alpha]$ is to be specified later. Fix some context $s$ and $v_1 \neq v_2$ different at entry $s$ and corresponding reward $r_1$ and $r_2$. Without loss of generality, we assume $r_1(s, \cdot) = (1/2 + \delta, 1/2 - \delta)$ and $r_2(s, \cdot) = (1/2 - \delta, 1/2 + \delta)$. Then direct calculation implies that

$$\pi_1^*(\cdot|s) = \frac{1}{2}\max\{0, \eta\alpha^{-1}(r_1(s, \cdot) - \lambda)\} = 0.5\eta\alpha^{-1}(r_1(s, \cdot) - \lambda),$$

$$\pi_2^*(\cdot|s) = \frac{1}{2}\max\{0, \eta\alpha^{-1}(r_2(s, \cdot) - \lambda)\} = 0.5\eta\alpha^{-1}(r_2(s, \cdot) - \lambda),$$

where $\lambda = 0.5 - \eta^{-1}\alpha$. Note that $2\chi^2(\mu\|\nu) + 1 = \sum_{a \in \mathcal{A}}[\mu(a)]^2/\nu(a)$ and $\chi^2 = D_f$, we obtain that $\forall \widehat{\pi}$,

$$\text{SubOpt}_s(\widehat{\pi}(\cdot|s); r_1) + \text{SubOpt}_s(\widehat{\pi}(\cdot|s); r_2) \tag{E.4}$$

$$= \langle r_1(s, \cdot), \pi_1^*(\cdot|s)\rangle + \langle r_2(s, \cdot), \pi_2^*(\cdot|s)\rangle - \overbrace{\langle r_1(s, \cdot) + r_2(s, \cdot), \widehat{\pi}(\cdot|s)\rangle}^{=1} + \overbrace{2\eta^{-1}\alpha\chi^2(\widehat{\pi}(\cdot|s)\|\pi^{\text{ref}}(\cdot|s))}^{\geq 0}$$

$$- \eta^{-1}\alpha \cdot \chi^2(\pi_1^*(\cdot|s)\|\pi^{\text{ref}}(\cdot|s)) - \eta^{-1}\alpha \cdot \chi^2(\pi_2^*(\cdot|s)\|\pi^{\text{ref}}(\cdot|s))$$

$$\geq 2\langle r_1(s, \cdot), \pi_1^*(\cdot|s)\rangle - 1 - 2\eta^{-1}\alpha \cdot \chi^2(\pi_1^*(\cdot|s)\|\pi^{\text{ref}}(\cdot|s))$$

$$= 1 + \frac{2\eta\delta^2}{\alpha} - 1 - \frac{\eta\delta^2}{\alpha} = \frac{\eta\delta^2}{\alpha}. \tag{E.5}$$

Now we take expectation over all possible contexts and recall that $\|v - v'\|_1 \geq S/2$ for $v \neq v'$, we know that for any $r_1 \neq r_2 \in \mathcal{G}$

$$\text{SubOpt}(\widehat{\pi}; r_1) + \text{SubOpt}(\widehat{\pi}; r_2) \geq \frac{\eta\delta^2}{2\alpha}$$

Given any mean reward function $r \in \mathcal{G}$, let $P_r$ be the distribution of $(s, a, \mathbf{r})$ when $s \sim \rho$, $a \sim \pi^{\text{ref}}(\cdot|s)$, and $\mathbf{r} \sim \text{Bern}(r(s, a))$. Suppose $P_{\mathcal{D}_r}$ is the distribution of the dataset given mean reward function $r$, then $\text{KL}\left(P_{\mathcal{D}_{r_1}}\|P_{\mathcal{D}_{r_2}}\right) = n\text{KL}\left(P_{r_1}\|P_{r_2}\right)$ for any pair of $r_1, r_2 \in \mathcal{G}$. Now we invoke Fano's inequality (Lemma H.2) to obtain

$$\inf_{\pi} \sup_{\text{inst} \in \text{CB}_\mathcal{G}} \text{SubOpt}(\widehat{\pi}; \text{inst}) \geq \frac{\eta\delta^2}{4\alpha}\left(1 - \frac{\max_{r_1 \neq r_2 \in \mathcal{G}} \text{KL}\left(P_{\mathcal{D}_{r_1}}\|P_{\mathcal{D}_{r_2}}\right) + \log 2}{\log |\mathcal{G}|}\right)$$

$$\geq \frac{\eta\delta^2}{4\alpha}\left(1 - \frac{64n\delta^2 + 8\log 2}{S}\right),$$

where the second inequality holds due to $\text{KL}(\text{Bern}(p)\|\text{Bern}(q)) \leq (p - q)^2/[q(1 - q)]$. Let $\delta = 16^{-1}\sqrt{Sn^{-1}}$, then we obtain that for all $\pi$ we have

$$\sup_{\text{inst} \in \text{CB}_\mathcal{G}} \text{SubOpt}(\widehat{\pi}; \text{inst}) \gtrsim \frac{\eta S}{\alpha n},$$

which finishes the proof in that $\log_2|\mathcal{G}| = S$. $\qquad\square$

## F  GENERALIZATION TO CONTEXTUAL DUELING BANDITS

In this section, we extend our algorithm to the problems of regularized contextual dueling bandits, where the learner receives preference comparison instead of absolute signals. Our setup largely follows Zhu et al. (2023); Zhan et al. (2023) and the notion of sub-optimality follows Xiong et al. (2024); Zhao et al. (2024).

---

**Algorithm 3** Offline KL-Regularized Pessimistic Contextual Dueling Bandit (KL-PCDB)

---

**Require:** regularization $\eta$, reference policy $\pi^{\text{ref}}$, function class $\mathcal{G}$, offline dataset $\mathcal{D} = \{(s_i, a_i^1, a_i^2, y_i)\}_{i=1}^n$
  1: Compute the maximum likelihood estimator of the reward function

$$\bar{g} = \operatorname*{argmin}_{g \in \mathcal{G}} \sum_{i=1}^n \Big[ y_i \log \sigma\Big( \big[ g(s_i, a_i^1) - g(s_i, a_i^2) \big] \Big) + (1 - y_i) \log \sigma\Big( \big[ g(s_i, a_i^2) - g(s_i, a_i^1) \big] \Big) \Big]$$

  2: Let $\widehat{g}(s, a) = \bar{g}(s, a) - \Gamma_n(s, a)$, where $\Gamma_n(s, a)$ is the bonus term in (F.1)
**Ensure:** $\widehat{\pi}(a|s) \propto \pi^{\text{ref}}(a|s) \exp\big( \eta \cdot \widehat{g}(s, a) \big)$

---

### F.1 PROBLEM SETUP

We still consider contextual bandits $(\mathcal{S}, \mathcal{A}, r, \pi^{\text{ref}})$ where $\mathcal{S}$ is the state space, $\mathcal{A}$ is the action space and $r : \mathcal{S} \times \mathcal{A} \to [0, 1]$ is the reward function.[6] But only relative preference feedback is available, viz., we have an i.i.d. offline dataset $\mathcal{D} = \{(s_i, a_i^1, a_i^2, y_i)\}_{i=1}^n$, where $s_i \in \mathcal{S}$ is generated from distribution $\rho$ and $a_i^1, a_i^2 \sim \pi^{\text{ref}}$. The binary preference label $y_i = 1$ indicates $a_i^1$ is preferred over $a_i^2$ (denoted by $a^1 \succ a^2$) and 0 for $a^2 \succ a^1$ given context $s$. In this work we consider the Bradley-Terry Model, where $\mathbb{P}[y = 1 | s, a^1, a^2] = \sigma(r(s_i, a_i^1) - r(s_i, a_i^2))$, where $\sigma(x) = (1 + e^{-x})^{-1}$ is the link function. The objective here identical to (2.1) for KL-regularization and (3.1) for $f$-divergence regularization. Our goal is still to find an $\epsilon$-optimal policy. To control the complexity of the function class $\mathcal{G}$, we assume that Assumption 2.1 still holds here.

**Concentrability.** Analogous to Section 2, we need our estimation from offline dataset generalizable to the state-action pairs visited by our obtained policy. While density-ratio-based concentrability can be directly adapted to dueling bandit, we need a slightly different notion of $D^2$-divergence. This is because in dueling bandit, we cannot observe the absolute reward and best estimation $g$ we can achieve is that for any state $s$ and actions $a^1, a^2$, our estimated $g(s, a^1) - g(s, a^2) \approx r(s, a^1) - r(s, a^2)$. This implies that there exists some mapping $b : \mathcal{S} \to [-1, 1]$ such that $g(s, a) - b(s) \approx r(s, a)$ on the offline data, which leads to the following definition.

**Definition F.1.** Given a class of functions $\mathcal{G} \subset (\mathcal{S} \times \mathcal{A} \to \mathbb{R})$ and some policy $\pi$, let $\mathcal{B} = (\mathcal{S} \to [-1, 1])$ be the function class, define the $D^2$-divergence $D_{\mathcal{G}}^2((s, a); \pi)$ as

$$\sup_{g, h \in \mathcal{G}} \inf_{b \in \mathcal{B}} \frac{\big( g(s, a) - h(s, a) - b(s) \big)^2}{\mathbb{E}_{s \sim \rho} \operatorname{Var}_{a' \sim \pi(\cdot|s')}[g(s', a') - h(s', a')]}.$$

A similar definition has been introduced in Zhao et al. (2024, Definition 2.6), which underpins the following two assumptions that characterize the coverage ability of $\pi^{\text{ref}}$ similarly as in Section 2.

Given a reference policy $\pi^{\text{ref}}$, we define two coverage notions for contextual dueling bandits.

**Assumption F.2** (All-policy concentrability). $D^2 := \sup_{(s,a) \in \mathcal{S} \times \mathcal{A}} D_{\mathcal{G}}^2((s, a); \pi^{\text{ref}}) < \infty$.

**Assumption F.3** (Single-policy concentrability). $D_{\pi^*}^2 := \mathbb{E}_{(s,a) \sim \rho \times \pi^*}[D_{\mathcal{G}}^2((s, a); \pi^{\text{ref}})] < \infty$.

Similar single-policy concentrability assumptions have appeared in previous work in offline contextual dueling bandits (Huang et al., 2025b; Song et al., 2024) and similar notions has also appeared in the analysis of model-based RL (Uehara & Sun, 2021; Wang et al., 2024). Still, while Assumption F.3 is strictly weaker than Assumption F.2, in general cases, the two quantities, $C^{\pi^*}$ and $D_{\pi^*}^2$ cannot be bounded by each other.

### F.2 ALGORITHMS AND RESULTS

#### F.2.1 ALGORITHMS FOR KL-REGULARIZED CONTEXTUAL DUELING BANDITS

We elucidate KL-PCDB for offline KL-regularized contextual dueling bandits, whose pseudocode is summarized in Algorithm 3. KL-PCDB first estimate the ground truth function $g^*$ on offline dataset

---

[6]We overload some notations in Section 2 by their dueling counterparts for notational simplicity.

---

**Algorithm 4** Offline $f$-Divergence Regularized Contextual Dueling Bandits ($f$-CDB)

---

**Require:** regularization $\eta$, reference policy $\pi^{\text{ref}}$, function class $\mathcal{G}$, offline dataset $\mathcal{D} = \{(s_i, a_i^1, a_i^2, y_i)\}_{i=1}^n$

1: Compute the maximum likelihood estimator of the reward function

$$\bar{g} = \underset{g \in \mathcal{G}}{\arg\min} \sum_{i=1}^n \Big[ y_i \log \sigma \Big( \big[ g(s_i, a_i^1) - g(s_i, a_i^2) \big] \Big) + (1 - y_i) \log \sigma \Big( \big[ g(s_i, a_i^2) - g(s_i, a_i^1) \big] \Big) \Big].$$

2: Compute the optimal policy with respect to reward $\bar{g}$

$$\widehat{\pi}(\cdot|s) \leftarrow \underset{\pi(\cdot|s) \in \Delta(\mathcal{A})}{\arg\max} \sum_{a \in \mathcal{A}} \pi(a|s)\bar{g}(s, a) + \eta^{-1} D_f \big( \pi(\cdot|s) \| \pi^{\text{ref}}(\cdot|s) \big)$$

**Ensure:** $\widehat{\pi}(a|s)$

---

with maximum likelihood estimator (MLE) to estimate a function $\bar{g} \in \mathcal{G}$. After that, analogous to Algorithm 1, we adopt the principle of pessimism in the face of uncertainty. Specifically, we define the penalty term

$$\Gamma_n(s, a) = \beta \sqrt{D_{\mathcal{G}}^2((s, a), \pi^{\text{ref}})}, \tag{F.1}$$

where

$$\beta^2 = 128 \log(2\mathcal{N}_{\mathcal{G}}(\epsilon_c)/\delta)/3n + 18\epsilon_c = \widetilde{O}(n^{-1}) \tag{F.2}$$

and then subtract it from the MLE $\bar{g}$ to obtain a pessimistic estimator $\widehat{g}$. KL-PCB then output the policy $\widehat{\pi}$, maximizing the estimated objective

$$\widehat{J}(\pi) = \mathbb{E}_{(s,a) \sim \rho \times \pi} \left[ \widehat{g}(s, a) - \eta^{-1} \log \frac{\pi(a|s)}{\pi^{\text{ref}}(a|s)} \right],$$

the maximizer of which is in closed form as the counterpart of (2.2).

$$\widehat{\pi}(a|s) \propto \pi^{\text{ref}}(a|s) \exp \big( \eta \cdot \widehat{g}(s, a) \big).$$

We provide the following theoretical guarantees for Algorithm 3.

**Theorem F.4.** Under Assumption F.3, if we set $\Gamma_n$ according to (F.1), then for sufficiently small $\epsilon \in (0, 1)$, with probability at least $1 - \delta$, $n = \widetilde{O}\big(\eta(D_{\pi^*}^2 \wedge C^{\pi^*})\epsilon^{-1}\big)$ is sufficient to guarantee the output policy $\widehat{\pi}$ of Algorithm 3 to be $\epsilon$-optimal.

**Remark F.5.** Zhao et al. (2024) achieved an $\widetilde{O}(\epsilon^{-1})$ sample complexity under Assumption F.2. Comparing to Zhao et al. (2024), KL-PCDB achieves the same $\widetilde{O}(\epsilon^{-1})$ sample complexity but only requiring Assumption F.3, which is weaker than Assumption F.2.

The following theorem provides the sample complexity lower bound for KL-regularized dueling contextual bandits.

**Theorem F.6.** For any sufficiently small $\epsilon \in (0, 1), \eta > 0, 1 \leq C^* \leq \exp(\eta/2)/2$, and any algorithm Alg, there is a KL-regularized contextual dueling bandit instance with single-policy concentrability $C^{\pi^*} \leq C^*$ such that Alg requires at least $\Omega\big(\min\{\eta C^* \log \mathcal{N}_{\mathcal{G}}(\epsilon_c)/\epsilon, \log \mathcal{N}_{\mathcal{G}}(\epsilon_c)(C^*)^2/\epsilon^2\}\big)$ samples to return an $\epsilon$-optimal policy.

**Remark F.7.** Theorem F.6 shows that when $\epsilon$ is sufficiently small, any algorithm for offline KL-regularized contextual dueling bandits requires at least $\Omega(\eta C^{\pi^*} \log \mathcal{N}_{\mathcal{G}}(\epsilon)\epsilon^{-1})$ samples to output an $\epsilon$-optimal policy, which matches the sample complexity upper bound in Theorem F.4, indicating that KL-PCB is nearly optimal.

### F.2.2 ALGORITHM AND RESULTS FOR $f$-DIVERGENCE REGULARIZED CDBS

We present an offline learning algorithm for $f$-divergence regularized contextual dueling bandit, $f$-CDB, in Algorithm 4. $f$-CDB first leverages maximum likelihood estimator to find a function

$\bar{g} \in \mathcal{G}$ that minimizes its risk on the offline dataset. Then the algorithm constructs the output policy $\widehat{\pi}$ that maximizes the f-divergence regularized objective induced by $\bar{g}$. Similar to Algorithm 2, we do not require any pessimism in $f$-CDB. The following theorem provides an upper bound of Algorithm 4.

**Theorem F.8.** For any sufficiently small $\epsilon \in (0, 1)$, and $\eta, \alpha > 0$, with probability at least $1 - \delta$, $n = \widetilde{O}(\alpha^{-1}\eta \log \mathcal{N}(\epsilon)\epsilon^{-1})$ is sufficient to guarantee that the output policy $\widehat{\pi}$ of Algorithm 4 is $\epsilon$-optimal.

The following theorem provides a lower bound for offline $f$-divergence regularized contextual dueling bandit with strongly convex $f$.

**Theorem F.9.** For any $\epsilon \in (0, 1)$, $\alpha, \eta > 0$, and offline RL algorithm Alg, there is an $\alpha$-strongly convex $f$ and $f$-divergence regularized contextual dueling bandit instance such that Alg requires at least $\Omega(\alpha^{-1}\eta \log \mathcal{N}(\epsilon)\epsilon^{-1})$ samples to return an $\epsilon$-optimal policy.

**Remark F.10.** Theorem F.9 indicates that, when $\epsilon$ is sufficiently small, to produce an $\epsilon$-optimal policy, any algorithm for offline $f$-regularized contextual bandits with strongly convex $f$ requires at least $\widetilde{\Omega}(\alpha^{-1}\eta\epsilon^{-1})$ samples. This lower bound matches the sample-complexity upper bound in Theorem F.8, indicating that Algorithm 4 is nearly optimal.

# G  MISSING PROOF FROM APPENDIX F

## G.1  PROOF OF THEOREM F.4

The proof follows the proof in Section 2. At the beginning, we first define the event $\mathcal{E}(\delta)$ given $\delta > 0$ as

$$\mathcal{E}(\delta) := \left\{ \exists\, b : \mathcal{S} \to [-1, 1], \forall (s, a) \in \mathcal{S} \times \mathcal{A}, \left| \bar{g}(s, a) - b(s) - g^*(s, a) \right| \leq \Gamma_n(s, a) \right\}. \quad \text{(G.1)}$$

Here, $\Gamma_n$ is defined in (F.1). We abuse the notation and define $b(\cdot)$ as

$$b = \underset{\mathcal{B}}{\arg\min} \sup_{(s,a) \in \mathcal{S} \times \mathcal{A}} \Phi_b(s, a) - \Gamma_n(s, a), \quad \text{(G.2)}$$

where $\Phi_b(s, a) = \left| \bar{g}(s, a) - b(s) - g^*(s, a) \right|$ and when $\mathcal{E}$ holds, for all $(s, a) \in \mathcal{S} \times \mathcal{A}$, we have $\Phi_b(s, a) \leq \Gamma_n(s, a)$ This indicates that the least square estimation $\bar{g}$ obtained in Line 1 of Algorithm 3, after adjusted by some bias function $b$, is close to the true function $g^*$. The following lemma shows that this event holds with high probability.

**Lemma G.1.** For any $\delta > 0$, $\mathbb{P}(\mathcal{E}(\delta)) \geq 1 - \delta$.

*Proof.* From Lemma H.1, we have that with probability at least $1 - \delta$, it holds that

$$\mathbb{E}_{s' \sim \rho} \text{Var}_{a' \sim \pi^{\text{ref}}(\cdot|s')} \left[ \bar{g}(s', a') - g^*(s', a') \right] \leq O\left( \frac{1}{n} \log(\mathcal{N}_{\mathcal{G}}(\epsilon_c)/\delta) + \epsilon_c \right). \quad \text{(G.3)}$$

It further holds true that for some $b : \mathcal{S} \to \mathbb{R}$

$$D_{\mathcal{G}}^2((s, a), \pi^{\text{ref}})) \cdot \mathbb{E}_{s \sim \rho} \text{Var}_{a \sim \pi^{\text{ref}}(\cdot|s)} \left[ \bar{g}(s, a) - g^*(s, a) \right] \geq \left( \bar{g}(s, a) - b(s) - g^*(s, a) \right)^2. \quad \text{(G.4)}$$

Substituting (G.3) into (G.4), we have

$$\inf_b \left( \bar{g}(s, a) - b(s) - g^*(s, a) \right)^2 \quad \text{(G.5)}$$

$$= \inf_b \frac{\left( \bar{g}(s, a) - b(s) - g^*(s, a) \right)^2}{\mathbb{E}_{s' \sim \rho} \text{Var}_{a' \sim \pi^{\text{ref}}(\cdot|s')} \left[ \bar{g}(s', a') - g^*(s', a') \right]} \mathbb{E}_{s' \sim \rho} \text{Var}_{a' \sim \pi^{\text{ref}}(\cdot|s')} \left[ \bar{g}(s', a') - g^*(s', a') \right]$$

$$\leq D_{\mathcal{G}}^2((s, a), \pi^{\text{ref}}) \mathbb{E}_{\pi^{\text{ref}}} \left[ \left( \bar{g}(s, a) - b(s) - g^*(s, a) \right)^2 \right] \quad \text{(G.6)}$$

$$\leq D_{\mathcal{G}}^2((s, a), \pi^{\text{ref}}) O\left( \frac{1}{n} \log(\mathcal{N}_{\mathcal{G}}(\epsilon_c)/\delta) + \epsilon_c \right), \quad \text{(G.7)}$$

where the first inequality holds due to the definition of $D_{\mathcal{G}}^2((s, a), \pi^{\text{ref}})$ and the last inequality holds due to Lemma H.1. $\square$

We overload the following quantities. For any $\gamma \in [0, 1]$ and $(s, a) \in \mathcal{S} \times \mathcal{A}$, we define

$$g_\gamma(s, a) := \gamma(\widehat{g}(s, a) - b(s)) + (1 - \gamma)g^*(s, a).$$

Furthermore, we introduce the following quantities

$$\pi_\gamma(\cdot|\cdot) = \pi_{g_\gamma}(\cdot|\cdot) \propto \pi^{\mathsf{ref}}(\cdot|\cdot) \exp\left(\eta g_\gamma(\cdot, \cdot)\right),$$
$$G(\gamma) := \mathbb{E}_{\rho \times \pi_\gamma}\left[\left(\widehat{g}(s, a) - b(s) - g^*(s, a)\right)^2\right],$$

where $b(\cdot)$ is defined in (G.2). We still have the monotonicity of the function $G(\gamma)$, which is characterized by the following lemma.

**Lemma G.2.** On event $\mathcal{E}(\delta)$, $0 \in \operatorname{argmax}_{\gamma \in [0,1]} G(\gamma)$.

*Proof.* For simplicity, we use $\triangle(s, a)$ to denote $\widehat{g}(s, a) - b(s) - g^*(s, a)$ in *this* proof. Then on event $\mathcal{E}(\delta)$, we know that $\triangle(s, a) \leq 0$ for all $(s, a) \in \mathcal{S} \times \mathcal{A}$. Taking derivatives of $G$ w.r.t., $\gamma$ directly, we conclude that for all $\gamma \in [0, 1]$,

$$G'(\gamma) = \eta \mathbb{E}_\rho \mathbb{E}_{a \sim \pi_\gamma}\left[\triangle^2(s, a)\left(\triangle(s, a) - \mathbb{E}_{a' \sim \pi_\gamma}[\triangle(s, a')]\right)\right]$$
$$= \eta \mathbb{E}_\rho\left[\mathbb{E}_{\pi_\gamma}\left[\triangle^3(s, a)\right] - \mathbb{E}_{\pi_\gamma}\left[\triangle^2(s, a)\right]\mathbb{E}_{\pi_\gamma}\left[\triangle(s, a)\right]\right]$$
$$\leq 0,$$

where $\mathbb{E}_\rho$ is the shorthand of $\mathbb{E}_{s \sim \rho}$, $\mathbb{E}_{\pi_\gamma}$ is the shorthand of $\mathbb{E}_{a \sim \pi_\gamma}$ and the inequality holds conditioned on the event $\mathcal{E}(\delta)$ due to Lemma 2.15. $\qquad\square$

Finally, we have the proposition that adding some bias term $b : \mathcal{S} \to \mathbb{R}$ does not affect the resulting policy.

**Proposition G.3.** Let $b : \mathcal{S} \to \mathbb{R}$ be some bias function, then for all $g \in \mathcal{G}$ we have $J(\pi_g) = J(\pi_{g-b})$, where $(g - b)(s, a) = g(s, a) - b(s)$.

*Proof.* For any fixed state $s \in \mathcal{S}$, we have for any $a \in \mathcal{A}$ that,

$$\pi_g(a|s) = \frac{\pi^{\mathsf{ref}}(a|s) \exp\left(\eta g(s, a)\right)}{\sum_{a' \in \mathcal{A}} \pi^{\mathsf{ref}}(a'|s) \exp\left(\eta g(s, a')\right)}$$
$$= \frac{\pi^{\mathsf{ref}}(a|s) \exp\left(\eta g(s, a)\right) \exp\left(-\eta b(s)\right)}{\sum_{a' \in \mathcal{A}} \pi^{\mathsf{ref}}(a'|s) \exp\left(\eta g(s, a')\right) \exp\left(-\eta b(s)\right)}$$
$$= \frac{\pi^{\mathsf{ref}}(a|s) \exp\left(\eta[g(s, a) - b(s)]\right)}{\sum_{a' \in \mathcal{A}} \pi^{\mathsf{ref}}(a'|s) \exp\left(\eta[g(s, a') - b(s)]\right)}$$
$$= \pi_{g-b}(a|s),$$

which indicates that $\pi_g = \pi_{g-b}$. This immediately leads to $J(\pi_g) = J(\pi_{g-b})$. $\qquad\square$

Now we are ready to prove Theorem F.4.

*Proof of Theorem F.4.* We proceed the proof under the event $\mathcal{E}(\delta)$. By Proposition G.3, we know that

$$J(\pi^*) - J(\widehat{\pi}) = J(\pi^*) - J(\pi_{\widehat{g}})$$
$$= J(\pi^*) - J(\pi_{\widehat{g}-b}).$$

Consequently, there exist some $\gamma \in [0, 1]$ and $b : \mathcal{S} \to [-1, 1]$ such that

$$J(\pi^*) - J(\widehat{\pi}) = J(\pi^*) - J(\pi_{\widehat{g}-b})$$
$$\leq \eta \mathbb{E}_{\rho \times \pi_\gamma}\left[\left(\widehat{g}(s, a) - b(s) - g^*(s, a)\right)^2\right]$$
$$= \eta G(\gamma), \tag{G.8}$$

where the inequality holds due to Lemma 2.14. Under event $\mathcal{E}(\delta)$, we know that $\widehat{g}(s, a) - b(s) \leq g^*(s, a)$. Together with Lemma G.2, we obtain $G(\gamma) \leq G(0)$. Therefore, we know that

$$J(\pi^*) - J(\widehat{\pi}) \leq G(0) \tag{G.9}$$

$$= \eta \mathbb{E}_{\rho \times \pi^*}\left[\left(\widehat{g}(s, a) - b(s) - g^*(s, a)\right)^2\right]$$

$$\leq 4\eta\left(\mathbb{E}_{\rho \times \pi^*}\left[\Gamma_n^2(s, a)\right] \wedge C^{\pi^*} \mathbb{E}_{\rho \times \pi^*}\left[\left(\widehat{g}(s, a) - b(s) - g^*(s, a)\right)^2\right]\right)$$

$$= 4\eta\left(\beta^2 \mathbb{E}_{\rho \times \pi^*}\left[D_{\mathcal{G}}^2((s, a); \pi^{\mathsf{ref}})\right] \wedge C^{\pi^*} \mathbb{E}_{\rho \times \pi^*}\left[\left(\widehat{g}(s, a) - b(s) - g^*(s, a)\right)^2\right]\right)$$

$$= \widetilde{O}\left(\eta D_{\pi^*}^2 \log \mathcal{G}(\epsilon_c) n^{-1}\right), \tag{G.10}$$

where the inequality holds due to the definition of $\mathcal{E}(\delta)$. Plugging (G.10) into (G.8), we know that $J(\pi^*) - J(\widehat{\pi})$ has upper bound $\widetilde{O}(D_{\pi^*}^2 n^{-1})$. By Lemma G.1, event $\mathcal{E}$ with probability at least $1 - \delta$, which concludes the proof. □

### G.2 Proof of Theorem F.6

*Proof of Theorem F.6.* The proof is similar to the proof of Theorem 2.11. Consider the following family of contextual dueling bandit instances with $S := |\mathcal{S}|, A := |\mathcal{A}| < \infty$ and reward in some function class $\mathcal{G}$.

$$\mathrm{CDB} := \{(\mathcal{S}, \mathcal{A}, \rho, r, \pi^{\mathsf{ref}}, \eta) : r \in \mathcal{G}, \rho \in \Delta(\mathcal{S}), \pi^{\mathsf{ref}} \in \Delta(\mathcal{A}|\mathcal{S})\}. \tag{G.11}$$

Fixing any $S \geq 1, \eta > 4 \log 2$ and $C^* \in (2, \exp(\eta/4)]$, we aim to prove that, for any estimator $\mathcal{D} \mapsto \widehat{\pi} \in \Delta(\mathcal{A}|\mathcal{S})$, for any $n \geq 16SC^*$, there exist some function class $\mathcal{G}$, such that $\exists$ inst $= (\mathcal{S}, \mathcal{A}, \rho, r, \pi^{\mathsf{ref}}, \eta) \in \mathrm{CDB}$ with single-policy concentrability $C^{\pi^*} \leq C^*$, regularization coefficient $\eta, |\mathcal{S}| = S = \Theta(\log |\mathcal{G}|)$, and

$$\inf_{\mathrm{inst} \in \mathrm{CDB}} \mathrm{SubOpt}_{\mathrm{RKL}}(\widehat{\pi}; \mathrm{inst}) \gtrsim \min\{\eta SC^* n^{-1}, (SC^*)^{1/2} n^{-1/2}\}. \tag{G.12}$$

Since $\log |\mathcal{G}| \geq \log \mathcal{N}_{\mathcal{G}}(\epsilon)$ for any $\epsilon \in (0, 1)$, the above bound yields the desired result.

We construct the same reward function class as in the proof of Theorem 2.11. In particular, we set $\mathcal{S} = [S], \mathcal{A} = \{\pm 1\}, \rho = \mathsf{Unif}(\mathcal{S})$, and the reference policy to be

$$\forall s \in \mathcal{S}, \pi^{\mathsf{ref}}(-1|s) = C^{-1}, \pi^{\mathsf{ref}}(+1|s) = 1 - C^{-1};$$

where $C = C^*$. Then the total sub-optimality of any $\pi \in \Delta(\mathcal{A}|\mathcal{S})$ given any reward function $r : \mathcal{S} \times \mathcal{A} \to \mathbb{R}$ is

$$\mathrm{SubOpt}_{f\mathrm{div}}(\pi; r) = \frac{1}{S} \sum_{s=1}^{S} \mathrm{SubOpt}_{f\mathrm{div}}\left(\pi(\cdot|s); r(s, \cdot)\right). \tag{G.13}$$

We further let $\alpha = \eta^{-1} \log(C - 1) \Leftrightarrow C - 1 = \exp(\eta\alpha)$. We construct $2^S$ Bernoulli reward functions, in particular, $\forall \tau \in \{\pm 1\}^S$, the mean function $r_\tau$ of the reward (indexed by $\tau$) is defined as

$$r_\tau(s, -1) = 0.5 + \tau_s \delta, r_\tau(s, +1) = 0.5 - \alpha.$$

Then, following the derivation of (D.12) and (D.13), we know that $\forall s \in \mathcal{S}, \forall \tau, \tau' \in \{\pm 1\}^S$ with $\tau \sim_s \tau'$,

$$\mathrm{SubOpt}_s(\widehat{\pi}; \tau) + \mathrm{SubOpt}_s(\widehat{\pi}; \tau') \geq \frac{\eta\delta^2}{8} \wedge \frac{3\delta}{10}. \tag{G.14}$$

Let $P_r$ be the distribution of $(s, a^1, a^2, y)$ for $s \sim \rho, a^1, a^2 \overset{\mathrm{i.i.d.}}{\sim} \pi^{\mathsf{ref}}(\cdot|s)$ and $y \sim \mathsf{Bern}(\sigma(r(s, a^1) - r(s, a^2)))$. Now we set $\delta = \sqrt{S/n}$ and conclude that for $\tau \sim \tau'$ with $\tau_s = -\tau_s'$,

$\mathsf{KL}\left(P_{r_\tau} \| P_{r_{\tau'}}\right)$

$$= \frac{(C - 1)}{SC^2} \sum_{s', a^1, a^2} \mathsf{KL}\left(\mathsf{Bern}(\sigma(r_\tau(s', a^1) - r(s', a^2))) \| \mathsf{Bern}(\sigma(r_{\tau'}(s', a^1) - r(s', a^2)))\right)$$

$$= \frac{2(C - 1)}{SC^2}\left(\mathsf{KL}\left(\mathsf{Bern}(\sigma(\alpha + \delta)) \| \mathsf{Bern}(\sigma(\alpha - \delta))\right) \vee \mathsf{KL}\left(\mathsf{Bern}(\sigma(\alpha - \delta)) \| \mathsf{Bern}(\sigma(\alpha + \delta))\right)\right).$$

Since $\alpha, \delta \in (0, 1/2)$, by the fact $\mathsf{KL}(P\|Q) \leq 2Q_{\min}^{-1}\mathsf{TV}(P\|Q)^2$ (see e.g., Polyanskiy & Wu (2025, Section 7.6)), we know that

$$
\begin{aligned}
\mathsf{KL}\left(P_{r_\tau}\|P_{r_{\tau'}}\right) &\leq \frac{2(C-1)}{SC^2}\frac{4}{1+\exp(\alpha+\delta)}\left(\frac{1}{1+\exp(\alpha-\delta)} - \frac{1}{1+\exp(\alpha+\delta)}\right)^2 \\
&\leq \frac{4}{3SC}\frac{\exp(2\alpha)(\exp(\delta)-\exp(-\delta))^2}{(1+\exp(\alpha-\delta))^4} \\
&\leq \frac{4e}{3SC}(\exp(\delta)-\exp(-\delta))^2 \\
&\leq 36S^{-1}C^{-1}\delta^2,
\end{aligned} \tag{G.15}
$$

where the second and third inequality hold due to $\alpha, \delta \leq 1/2$, and last inequality follows from $\exp(x) - \exp(-x) \leq 3x$ for $x \in [0, 1/2]$. Now we set $\delta = \sqrt{SC/n} \leq 1/4$. We substitute (G.14) into Assouad's Lemma (Lemma H.3) and obtain that

$$
\begin{aligned}
\inf_{\text{inst}\in\text{CDB}}\text{SubOpt}_{\text{RKL}}(\widehat{\pi};\text{inst}) &\geq \frac{1}{4}S \cdot \frac{1}{S} \cdot \left(\frac{\eta\delta^2}{8} \wedge \frac{3\delta}{10}\right) \cdot \min_{\tau\sim\tau'}\exp\left(-\mathsf{KL}\left(P_{\mathcal{D}_\tau}\|P_{\mathcal{D}_{\tau'}}\right)\right) \\
&= \frac{1}{4}\left(\frac{\eta\delta^2}{8} \wedge \frac{3\delta}{10}\right)\exp\left(-n\mathsf{KL}\left(P_{r_\tau}\|P_{r_{\mathcal{D}_{\tau'}}}\right)\right) \\
&\geq \frac{\exp(-36)}{32}\min\{\eta CSn^{-1}, S^2C^2n^{-2}\},
\end{aligned}
$$

where the $1/S$ comes from the denominator of (G.13) and the second inequality follows from (G.15). $\qquad\square$

## G.3 Proof of Theorem F.8

*Proof of Theorem F.8.* The proof is similar to the proof of Theorem 3.2. Recall that $b(\cdot)$ defined in (G.2), we know that

$$
\begin{aligned}
\widehat{\pi} &= \operatorname*{argmax}_{\pi\in\Delta^d}\left\{\mathbb{E}_{(s,a)\sim\rho\times\pi}[\bar{g}(s,a)] - \eta^{-1}\mathbb{E}_{s\sim\rho}\left[D_f(\pi\|\pi^{\text{ref}})\right]\right\} \\
&= \operatorname*{argmax}_{\pi\in\Delta^d}\left\{\mathbb{E}_{(s,a)\sim\rho\times\pi}[\bar{g}(s,a) - b(s)] - \eta^{-1}\mathbb{E}_{s\sim\rho}\left[D_f(\pi\|\pi^{\text{ref}})\right]\right\}.
\end{aligned}
$$

We have the following sub-optimality decomposition

$$
\begin{aligned}
J(\pi^*) - J(\widehat{\pi}) &= \mathbb{E}_{s\sim\rho}\left[\mathbb{E}_{a\sim\pi^*}[g^*(s,a)] - \mathbb{E}_{a\sim\widehat{\pi}}[g^*(s,a)] - \eta^{-1}\left[D_f(\pi^*\|\pi^{\text{ref}}) - D_f(\widehat{\pi}\|\pi^{\text{ref}})\right]\right] \\
&= \mathbb{E}_{s\sim\rho}\left[H_s^*(g^*) - H_s^*(\bar{g} - b) - \langle\widehat{\pi}, g^* - \bar{g} + b\rangle\right] \\
&= \mathbb{E}_{s\sim\rho}\left[H_s^*(g^*) - H_s^*(\bar{g} - b) - \langle\nabla H_s^*(\bar{g} - b), g^* - \bar{g} + b\rangle\right] \\
&= \mathbb{E}_{s\sim\rho}[(g^* - \bar{g} + b)^\top\nabla^2 H_s^*(\widetilde{g})(g^* - \bar{g} + b)],
\end{aligned}
$$

where $\widetilde{g} = \gamma g^* + (1-\gamma)\bar{g}$ and $\gamma \in [0, 1]$, $(\bar{g} - b)(s, a) = \bar{g}(s, a) - b(s)$ and the last equation holds due to Taylor's expansion. Now, for any $\delta \in (0, 1)$ and $\epsilon_c > 0$, with probability at least $1 - \delta$

$$
\begin{aligned}
J(\pi^*) - J(\widehat{\pi}) &= \mathbb{E}_{s\sim\rho}[(g^* - \bar{g} + b)^\top\nabla^2 H_s^*(\widetilde{g})(g^* - \bar{g} + b)] \\
&\leq \alpha^{-1}\eta\mathbb{E}_{s\sim\rho}\left[(g^* - \bar{g} + b)^\top\text{diag}\left(\pi^{\text{ref}}(a_1|s), \cdots, \pi^{\text{ref}}(a_d|s)\right)(g^* - \bar{g} + b)\right] \\
&= \alpha^{-1}\eta\mathbb{E}_{(s,a)\sim\rho\times\pi^{\text{ref}}}\left[\left(g^*(s,a) - \bar{g}(s,a) + b(s)\right)^2\right] \\
&\leq \alpha^{-1}\eta\left(\frac{128}{3n}\log(2\mathcal{N}_{\mathcal{G}}(\epsilon_c)/\delta) + 18\epsilon_c\right),
\end{aligned}
$$

where the first inequality holds due to Lemma E.4 and last inequality holds due to equation (G.3). Setting $\epsilon_c = O(n^{-1})$ completes the proof. $\qquad\square$

## G.4 PROOF OF THEOREM F.9

*Proof of Theorem F.9.* We still consider the contextual dueling bandit instance class defined in (G.11). We show that given any positive $\alpha, \eta$, for any $n \geq S \cdot \max\{16, \eta^2\alpha^{-2}\}$, there exists $f : \mathbb{R} \to \mathbb{R}$ such that $f$ is $\alpha$-strongly convex, $\log|\mathcal{G}| = \Theta(S)$ and

$$\inf_{\widehat{\pi}\in\widehat{\Pi}(\mathcal{D})} \sup_{\text{inst}\in\text{CDB}} \text{SubOpt}_{f\text{div}}(\widehat{\pi}; \text{inst}) \gtrsim \frac{\eta S}{\alpha n}, \tag{G.16}$$

where $\mathcal{D} = \{(s_i, a_i^1, a_i^2, y_i)\}_{i=1}^n$ is the offline preference dataset, all (possibly randomized) maps from which to $\Delta(\mathcal{A}|\mathcal{S})$ is denoted by $\widehat{\Pi}(\mathcal{D})$. Since $S = \Theta(\log|\mathcal{G}|) \gtrsim \log\mathcal{N}_{\mathcal{G}}(\epsilon_c)$ for all $\epsilon_c \in (0, 1)$, we can conclude the theorem.

Let $\mathcal{S} = [S]$, $\mathcal{A} = \{\pm1\}$, $\rho = \text{Unif}(\mathcal{S})$ and $\pi^{\text{ref}}(\cdot|s) = \text{Unif}(\mathcal{A})$ for any $s \in \mathcal{S}$. Then the total sub-optimality of any $\pi \in \Delta(\mathcal{A}|\mathcal{S})$ given any reward function $r : \mathcal{S} \times \mathcal{A} \to \mathbb{R}$ is

$$\text{SubOpt}_{f\text{div}}(\pi; r) = \frac{1}{S}\sum_{s=1}^S \text{SubOpt}_{f\text{div}}(\pi(\cdot|s); r(s, \cdot)). \tag{G.17}$$

We still consider the reward function class $\mathcal{G}$ indexed by $\{\pm1\}^\mathcal{S}$. For all $\tau \in \{\pm1\}^\mathcal{S}$ the reward instance "shaped" by $\tau$ is

$$r_\tau(s, a) = \frac{1}{2} + a\tau_s \cdot \sqrt{\frac{S}{n}}, \tag{G.18}$$

where $a\tau_s = \pm1$ because $a \in \mathcal{A} = \{\pm1\}$. We thereby refer $\tau \sim \tau'$ to any pair in $\{\pm1\}^\mathcal{S}$ that differs only in one coordinate. $\forall \tau, \tau' \in \{\pm1\}^\mathcal{S}$, if $\tau \sim \tau'$, then suppose $\tau_s = -\tau'_s$, we have

$$\text{SubOpt}_{f\text{div}}(\pi(\cdot|s); r_\tau(s, \cdot)) + \text{SubOpt}_{f\text{div}}(\pi(\cdot|s); r_{\tau'}(s, \cdot)) \geq \frac{\eta S}{\alpha n}, \tag{G.19}$$

where the inequality follows from exactly the same calculation in equation (E.5) by setting $f(x) = \alpha(x-1)^2/2$.[7] Let $P_r$ be the distribution of $(s, a^1, a^2, y)$ for $s \sim \rho$, $a^1, a^2 \overset{\text{i.i.d.}}{\sim} \pi^{\text{ref}}(\cdot|s)$ and $y \sim \text{Bern}(\sigma(r(s, a^1) - r(s, a^2)))$. Then we denote $\delta = \sqrt{S/n}$ and conclude that for $\tau \sim \tau'$ with $\tau_s = -\tau'_s$,

$$\text{KL}\left(P_{r_\tau}\|P_{r_{\tau'}}\right) = \frac{1}{SA^2}\sum_{s',a^1,a^2} \text{KL}\left(\text{Bern}(\sigma(r_\tau(s', a^1) - r(s', a^2)))\|\text{Bern}(\sigma(r_{\tau'}(s', a^1) - r(s', a^2)))\right)$$

$$= \frac{1}{4S}\left(\text{KL}\left(\text{Bern}(\sigma(2\delta))\|\text{Bern}(\sigma(-2\delta))\right) + \text{KL}\left(\text{Bern}(\sigma(-2\delta))\|\text{Bern}(\sigma(2\delta))\right)\right)$$

$$\leq \frac{1}{4S}\left((\exp(-2\delta) - 1)^2 + (\exp(2\delta) - 1)^2\right)$$

$$\leq \frac{1}{2S}(\exp(2\delta) - 1)^2 \leq \frac{36\delta^2}{2S} = \frac{18}{n}, \tag{G.20}$$

where the last inequality follows from $\exp(x) - 1 \leq 3x$ for $x \in [0, 0.5]$ and $\delta = \sqrt{S/n} \leq 0.25$ by assumption. Therefore, we substitute (G.19) into Assouad's Lemma (Lemma H.3) to obtain

$$\text{LHS of (G.16)} \geq \frac{1}{S} \cdot S \cdot \frac{\eta S}{\alpha n} \cdot \frac{1}{4} \cdot \min_{\tau\sim\tau'} \exp\left(-\text{KL}\left(P_{\mathcal{D}_\tau}\|P_{\mathcal{D}_{\tau'}}\right)\right)$$

$$= 0.25 \cdot \frac{\eta S}{\alpha n} \cdot \exp\left(-n\text{KL}\left(P_{r_\tau}\|P_{r_{\mathcal{D}_{\tau'}}}\right)\right) \geq \frac{\eta S}{\alpha n} \cdot \frac{1}{3} \cdot \exp(-18) \gtrsim \frac{\eta S}{\alpha n}, \tag{G.21}$$

where the $1/S$ comes from the denominator of (G.17) and the second inequality follows from (G.20). □

---

[7] Recall that in this case $D_f = \chi^2$, where $2\chi^2(\mu\|\nu) + 1 = \sum_{a\in\mathcal{A}}[\mu(a)]^2/\nu(a)$.

# H   AUXILIARY LEMMAS

**Lemma H.1** (Zhao et al. 2024, Lemma D.4). Consider a offline dataset $\{(s_i, a_i^1, a_i^2, y_i)\}_{i=1}^n$ generated from the product of the context distribution $\rho \in \Delta(\mathcal{S})$, policy $\pi \in \Delta(\mathcal{A}|\mathcal{S})$, and the Bradley-Terry Model defined in Appendix F.1. Suppose $\bar{g}$ is the result of MLE estimation of Algorithm 3, and we further define $b(s) = \mathbb{E}_{a \sim \pi(\cdot|s)}[\bar{g}(s,a) - g^*(s,a)]$, then with probability at least $1 - 2\delta$, we have

$$\mathbb{E}_{s,a \sim \rho \times \pi}\left[\left(\bar{g}(s,a) - g^*(s,a) - b(s)\right)^2\right] \leq O\left(\frac{1}{n}\log(\mathcal{N}_{\mathcal{G}}(\epsilon_c)/\delta) + \epsilon_c\right).$$

Lemmas H.2 and H.3 are two standard reductions (Le Cam, 1973; Yu, 1997; Polyanskiy & Wu, 2025). See, e.g., Chen et al. (2024, Section 3) for a general proof.

**Lemma H.2** (Fano's inequality). Fix any $\mathcal{R} := \{r_1, \cdots, r_S\}$ and policy class $\Pi$, let $L : \Pi \times \mathcal{R} \to \mathbb{R}_+$ be some loss function. Suppose there exist some constant $c > 0$ such that the following condition holds:

$$\min_{i \neq j} \min_{\pi \in \Pi} L(\pi, r_i) + L(\pi, r_j) \geq c.$$

Then we have

$$\inf_{\pi \in \Pi} \sup_{r \in \mathcal{R}} \mathbb{E}_{\mathcal{D} \sim P_r} L(\pi(\mathcal{D}), r) \geq \frac{c}{2}\left(1 - \frac{\max_{i \neq j} \mathsf{KL}(P_{r_i} \| P_{r_j}) + \log 2}{\log S}\right),$$

where the trajectory distribution of $\pi$ interacting with instance $r \in \mathcal{R}$ is denoted by $P_r$.

**Lemma H.3** (Assouad's Lemma). Let $\mathcal{R}$ be the set of instances, $\Pi$ be the set of estimators, $\Theta := \{\pm 1\}^S$ for some $S > 0$, and $\{L_j\}_{j=1}^S$ be $S$ functions from $\Pi \times \mathcal{R}$ to $\mathbb{R}_+$. Suppose $\{r_\theta\}_{\theta \in \Theta} \subset \mathcal{R}$ and the loss function is

$$L(\pi, r) := \sum_{j=1}^S L_j(\pi, r), \forall (\pi, r) \in \Pi \times \mathcal{R}.$$

We denote $\theta \sim_j \theta'$ if they differ only in the $j$-th coordinate. Further assume that

$$\theta \sim_j \theta' \Rightarrow \inf_{\pi \in \Pi} L_j(\pi, r_\theta) + L_j(\pi, r_{\theta'}) \geq c \tag{H.1}$$

for some $c > 0$, then

$$\inf_{\pi \in \Pi} \sup_{r \in \mathcal{R}} \mathbb{E}_{\mathcal{D} \sim P_r} L(\pi(\mathcal{D}), r) \geq S \cdot \frac{c}{4} \min_{\exists j: \theta \sim_j \theta'} \exp\left(-\mathsf{KL}\left(P_{r_\theta} \| P_{r_{\theta'}}\right)\right),$$

where the trajectory distribution of $\pi$ interacting with instance $r \in \mathcal{R}$ is denoted by $P_r$.

The following Lemma H.4 is due to Gilbert (1952); Varshamov (1957), which is a classical result in coding theory.

**Lemma H.4.** Suppose $\Sigma$ is a set of characters with $|\Sigma| = q$ where $q \geq 2$ is a prime power and $N > 0$ is some natural number. Then there exists a subset $\mathcal{V}$ of $\Sigma^N$ such that (1) for any $v, v' \in \mathcal{V}, v \neq v_j$, one has $d_H(v, v') \geq N/2$ and (2) $\log_q |\mathcal{V}| \geq H_q(1/2) = \Theta(1)$, where $d_H$ is the Hamming distance and the entropy function $H$ is given by

$$H_q(x) = x\frac{\log(q-1)}{\log q} - x\frac{\log x}{\log q} - (1-x)\frac{\log(1-x)}{\log q}.$$

For example, when $q = 2$, this means that there exists a subset $\mathcal{V}$ of $\{-1, 1\}^S$ such that (1) $|\mathcal{V}| \geq \exp(S/8)$ and (2) for any $v, v' \in \mathcal{V}, v \neq v_j$, one has $\|v - v'\|_1 \geq S/2$.

