# OpenReview forum: "Towards a Sharp Analysis of Offline Policy Learning for $f$-Divergence-Regularized Contextual Bandits"
_ICLR.cc/2026/Conference — ICLR 2026 Poster_

### Official Review · Reviewer_gwK9 · 2025-10-24

**Soundness:** 3
**Presentation:** 3
**Contribution:** 3
**Rating:** 6
**Confidence:** 4

**Summary:**

This paper provides a sharp theoretical analysis of offline contextual bandits under f-divergence regularization, a class of objectives underlying many modern offline RL and RLHF algorithms. The key contribution is a refined characterization of the sample complexity required to find an ε-optimal policy with respect to the regularized objective, rather than the unregularized reward.

For the reverse KL divergence, the authors prove an Ō(ε⁻¹) sample complexity under single-policy concentrability—strictly improving those requiring stronger all-policy coverage assumptions. They also provide a matching lower bound showing that the dependency on the single-policy concentrability coefficient is necessary.

For general f-divergences with strongly convex f, they establish that the same fast rate Θ̃(ε⁻¹) is achievable without any coverage dependence, supported by both upper and lower bounds. The analysis leverages a novel moment-based argument that couples pessimism with the curvature of the regularized objective, providing a new proof technique beyond the standard performance-difference or simulation-lemma analyses.

Empirical results on two-armed synthetic bandits corroborate the theoretical rates and demonstrate distinct scaling behaviors between KL and strongly convex f-divergences.

**Strengths:**

- Sharp and rigorous theoretical results: The paper achieves the first tight sample complexity bounds for offline f-divergence-regularized bandits, closing important gaps in the literature. The improvements—from stronger coverage assumptions to Ō(ε⁻¹) under single-policy coverage—are both technically and conceptually relevant.

- Broader generality beyond KL: Extending the framework to strongly convex f-divergences (e.g., χ²) and proving that fast rates can be achieved without any concentrability assumptions is conceptually strong. It clarifies how curvature of the regularizer directly impacts coverage requirements.

- Balanced theoretical and empirical validation: Although simple, the experiments convincingly verify the predicted n⁻¹ scaling and the absence of coverage dependence for strongly convex f. The theoretical lower bounds are also near-matching, strengthening the claims.

- Relevance and timeliness: The results have clear implications for RLHF, offline RL with KL or χ² regularization, and policy optimization under limited coverage, making the work both theoretically rigorous and practically relevant.

**Weaknesses:**

- The experiments only involve two-armed synthetic bandits. Although sufficient to verify scaling, a richer empirical study (e.g., linear contextual or synthetic high-dimensional bandits) would strengthen the practical credibility.
- Som part of the writting is unclear (see questions)

**Questions:**

- $\pi_{\gamma}$ in the LHF of the equation in Lemma 2.14?
- I don't understand Lemma 2.14, does the lemma hold for every $\lambda$, for for some $\lambda$ ($\exists \lambda$)?
- How critical is the assumption that the behavior policy equals π_ref? Would the same rates hold when π_ref differs from the logging policy?
- For the χ²-regularized case, could you provide intuition or experiments showing the trade-off between strong convexity (α) and the bias introduced by over-regularization?
- Could your analysis provide insights into why KL regularization remains dominant in practice despite its stronger coverage requirements?

---

> ### Author Response · Authors · 2025-11-21
>
> We thank the reviewer for the insightful and constructive feedback. We address the concerns as follows.
>
> **Q1**: The experiments only involve two-armed synthetic bandits.
>
> **A1**: Our original simulation on tabular hard instances is indeed a rigorous sanity check, validating the correctness of our theory. We further **add more experiments** for both KL and $f$ cases on synthetic linear bandits and **real-world datasets** in Section 4.2 and Section 4.3 of the revised submission.
>
> **Q2**: $\\pi\_{\\gamma}$ in the LHF of the equation in Lemma 2.14?
>
> **A2**: Thank you for pointing out this vagueness. Here, we mean that $g^\*$ is the ground-truth reward and $g$ be an estimated reward, then there exist a $\\gamma \\in [0,1]$ such that the inequality in Lemma 2.14 holds if we define $g\_{\\gamma} = \\gamma g + (1-\\gamma)g^\*$ and $\\pi\_{\\gamma} \\propto \\exp (\\eta g\_{\\gamma})$. We appologize for the confusion and have revised this part accordingly.
>
> **Q3**: I don't understand Lemma 2.14, does the lemma hold for every $\\lambda$, or for some $\\lambda$ ($\\exists \\lambda$)?
>
> **A3**: We believe the reviewer is referring to the $\\gamma$ in Lemma 2.14. Indeed, we mean there exists such a $\\gamma$ due to the mean-value theorem.

---

> > ### Author Response · Authors · 2025-11-21
> > **(Cont'd)**
> >
> > **Q4**: How critical is the assumption that the behavior policy equals π\_ref?
> >
> > **A4**: If the behavior policy $\\pi^\\mathrm{bhv}$ is different from $\\pi^\\mathrm{ref}$, the algorithm might not work (with respect to the $\\pi^{\\mathrm{ref}}$-regularized objective) in general because in this case the agent cannot infer the information of $\\pi^\\mathrm{ref}$ from the offline dataset alone, which means an additional (multiplicative) coverage coefficient between $\\pi^\\mathrm{bhv}$ and $\\pi^\\mathrm{ref}$ is expected to appear in both the upper and lower bounds. Therefore, a relaxation of this assumption may **only complicate the analysis but is still doable**. Moreover, this assumption is **standard** in literature [1,2,3] and quite **reasonable**, which is directly motivated by the empirical practice of using the rollouts from the reference policy (model) $\\pi^\\mathrm{ref}$ for further training and policy improvement in the context of LLM fine-tuning, in which $\\pi^\\mathrm{ref} \\leftarrow \\pi^\\mathrm{SFT}$ is often the policy obtained from the stage of supervised fine-tuning.
> >
> > **Q5**: For the χ²-regularized case, could you provide intuition or experiments showing the trade-off between strong convexity (α) and the bias introduced by over-regularization?
> >
> > **A5**: Note that $\\alpha$ always comes together with $\\eta$ in the form of $\\alpha / \\eta$ in our $f$-divergence regularized objective (3.1) because we can always devide the $f$-divergence by $\\alpha$ to normalize it into a $f$-divergence with $1$-strongly convex $f$. Here, $\\alpha / \\eta$ is intuitively the **effective temperature**, which corresponds to the $\\beta$ in [4, equation (13)]. Therefore, if the notion of sub-optimality is defined with respect to the **unregularized** objective, the bias introduced by over-regularization will be proportional to the effective temperature, which has been manifested in [4, Theorem 3.1]; however, note that our notion of sub-optimality is already defined via the **corresponding** $f$-divergence regularized objective, the bound in our Theorem 3.2 will no longer explicitly exhibit such a "over-regularization" term.
> >
> > **Q6**: Could your analysis provide insights into why KL regularization remains dominant in practice despite its stronger coverage requirements?
> >
> > **A6**: This is an intriguing question. In short, our analysis not only deepens the understanding of KL-regularization, but also suggests that $f$-divergence with strongly convex $f$ could also be good alternatives. In detail, we show that when $f$ is strongly convex, the corresponding objective is can be achieved in a coverage-free manner. This is primary because the regularization given by strongly convex $f$ forces the induced policy to remain sufficiently close to the behavior policy $\\pi\_{\\text{ref}}$. In contrast, KL-regularization is coverage dependent, allowing the induced policy deviating somewhat more from $\\pi\_{\\text{ref}}$. Therefore, KL-regularization might be more suitable in practice when one wishes to push output policy away from a suboptmal behavior policy. Also, KL-regularization can induce the closed Gibbs form in our Algorithm 1, which has already inspired some easy-to-implement methods for LLM fine-tuning, such as DPO [5]; and thus this algorithmic aspect may also serve as a reason for its popularity.
> >
> > References
> >
> > [1] Xie, Tengyang, et al. "Exploratory preference optimization: Harnessing implicit q*-approximation for sample-efficient rlhf." arXiv preprint arXiv:2405.21046 (2024).
> >
> > [2] Xiong, Wei, et al. "Iterative preference learning from human feedback: Bridging theory and practice for rlhf under kl-constraint." arXiv preprint arXiv:2312.11456 (2023).
> >
> > [3] Zhou, Xingyu, Yulian Wu, and Francesco Orabona. "A unified theoretical analysis of private and robust offline alignment: from rlhf to dpo." arXiv preprint arXiv:2505.15694 (2025).
> >
> > [4] Huang, Audrey, et al. "Correcting the mythos of kl-regularization: Direct alignment without overoptimization via chi-squared preference optimization." arXiv preprint arXiv:2407.13399 (2024).
> >
> > [5] Rafailov, Rafael, et al. "Direct preference optimization: Your language model is secretly a reward model." Advances in neural information processing systems 36 (2023): 53728-53741.

---

### Official Review · Reviewer_8Rei · 2025-11-01

**Soundness:** 3
**Presentation:** 3
**Contribution:** 3
**Rating:** 6
**Confidence:** 2

**Summary:**

This paper provides a tight statistical analysis of offline contextual bandits with f-divergence regularization, focusing on the widely used reverse KL and on general strongly convex f-divergences. The authors establish the $O(1/\epsilon)$ sample-complexity results under single-policy concentrability for reverse-KL-regularized objectives, A novel pessimism-based proof technique is used by leveraging the curvature of the reverse-KL objective and a moment-based lemma. For strongly convex f-divergences, the authors further show that the same rate can be achieved without any coverage assumptions, supported by a matching lower bound and simple empirical verification.

**Strengths:**

1. This paper established a near-optimal sample complexity for offline regularized bandits.
2. The moment-based argument and integration of pessimism with curvature properties seem novel in offline bandits.
3. The extension to strongly convex f-divergences provides a unified theoretical view.

**Weaknesses:**

1. Experiments are limited. Only toy two-armed-bandit cases are shown, perhaps including some real dataset would strengthen the arguments.
2. The paper seems to put more effort on introducing the analysis for KL-divergence regularized Bandits, while the title suggest general f-divergence. Perhaps including discussions on how to handle general f-divergence regularized bandits would make the main body match with the title.

**Questions:**

The paper mentioned that for strongly convex f-divergence, we do not need the data coverage assumption. I am wondering if we use the local convex f-divergence, can KL analysis be applied for this case? Or there is a fundamental challenge here.

---

> ### Author Response · Authors · 2025-11-21
>
> We thank the reviewer for the insightful and constructive feedback. We address the concerns as follows.
>
> **Q1**: Experiments are limited. Only toy two-armed-bandit cases are shown
>
> **A1**: Our original simulation on tabular hard instances is indeed a rigorous sanity check, validating the correctness of our theory. We further **add more experiments** for both KL and $f$ cases on synthetic linear bandits and **real-world datasets** in Section 4.2 and Section 4.3 of the revised submission.
>
> **Q2**: The paper seems to put more effort on introducing the analysis for KL-divergence regularized Bandits, while the title suggest general f-divergence.
>
> **A2**: Our two-fold contribution consists of two **disjoint** classes of regularizers, namely KL and $f$-divergence with strongly convex $f$. There are two reasons for giving KL a longer elaboration.
> 1. KL-regularization is arguably the mostly applied regularization and its statistical properties has also been investigated in previous works like [1,2]. KL divergence is also essentially the unique Bregman divergence that is also a $f$-divergence (with $f(x) = x\\log x$) [3]. These motivates a **tailored** analysis of KL-regularized objective.
> 2. Since the $f(x)=x\\log x$ for KL itself is **NOT** strongly convex, the regularization is not enough and therefore addtional algorithmic pessimism is required, which different from strongly convex $f$ case; note also that the elegant **(short)** upper bound **analysis** for the $f$ case is **enabled by the strong convexity of $f$**, which is a merit **absent in the KL case**! Finally, in terms of technical take-away, our matching bounds show the necessacity of coverage for KL-divergence, which also differs from $f$-divergence strongly convex $f$. We therefore separate and devote dedicated attention to KL-regularization.
>
> **Q3**: Can KL analysis be applied for locally convex $f$ ?
>
> **A3**: Currently our upper bound analysis for KL-regularized objectives, especially Lemma 2.14, indeed heavily hinges on the specific closed form solution $\\pi|\_{r} \\propto \\pi\_\\mathrm{ref}(\\cdot)\\exp(\\eta r(\\cdot))$, we leave a more general analysis for locally convex $f$, which is ideally free of the exploitation of the Gibbs form, as an important future direction.
>
>
> References:
>
> [1] Xiong, Wei, et al. "Iterative preference learning from human feedback: Bridging theory and practice for rlhf under kl-constraint." arXiv preprint arXiv:2312.11456 (2023).
>
> [2] Zhao, Heyang, et al. "Sharp analysis for kl-regularized contextual bandits and rlhf." arXiv preprint arXiv:2411.04625 (2024).
>
> [3] Jiao, Jiantao, et al. "Information measures: the curious case of the binary alphabet." IEEE Transactions on Information Theory 60.12 (2014): 7616-7626.

---

### Official Review · Reviewer_8QC9 · 2025-11-01

**Soundness:** 3
**Presentation:** 3
**Contribution:** 3
**Rating:** 6
**Confidence:** 2

**Summary:**

The paper investigates the theoretical offline reinforcement learning with general function approximation. They provide sharper lower and upper bounds with a proposed KL-divergence-regularized simple algorithm, where they also prove single-policy concentrability is efficient and necessary. Also, they investigate the f-divergence-regularized algorithm, which can converge normally without coverage assumptions. The results are very novel in this area. A few experiments are conducted to corroborate their theoretical results.

**Strengths:**

1. The paper is well-written. The story is clear and consistent.
2. The contributions should be solid. a) The f-divergence could provide another way to understand the requirement for offline learning. b) The refined mean-value-type risk upper bound has some technical improvements.

**Weaknesses:**

1. The results in the f-divergence algorithm are not compared with the global optimal policy, which is not the same as the KL-divergence. In other words, it seems to replace the coverage assumption with another assumption that the global optimal policy is the optimal policy they defined in 3.1. It is still an important contribution, but it may be overstated.
2. The numerical experiments are very simple, more like a sanity check rather than a validation.

**Questions:**

1. Could you provide more intuition for Lemma 2.14? It appears central to your KL-regularized algorithm. Also, this technique seems to originate in prior work—can you clarify the differences in proofs between this one and the previous one?
2. For the f-divergence regularized algorithm, is my understanding in the weakness correct? If no, please justify it. If yes, please elaborate a little more about what is the potential impact or benefit of it.

---

> ### Author Response · Authors · 2025-11-21
>
> We thank the reviewer for the insightful and constructive feedback. We address the questions as follows.
>
> **Q1**: The results in the f-divergence algorithm are not compared with the global optimal policy, which is not the same as the KL-divergence.
>
> **A1**: We thank the reviewer for acknowleding the significance of our results. Moreover, we have stated it clear in Section 3.1 that the $f$-divergence regularized objective with strongly convex $f$ is defined using this $f$-divergence. We would like to reiterate that our results for both settings together unveil the **distinct properties of different performance metrics**: when $\\diamondsuit$-divergence-regularized objective serves the performance metric, its optimal converence rate *in the worst case* **depends on** the single policy concentrability if $\\diamondsuit = \\mathrm{KL}$ but **does not depends on** any coverage condition if $\\diamondsuit = f$ with a strongly convex $f$.
>
> **Q2**: The numerical experiments are very simple, more like a sanity check rather than a validation
>
> **A2**: Our original simulation on tabular hard instances is indeed a rigorous sanity check, validating the correctness of our theory. We further **add more experiments** for both KL and $f$ cases on synthetic linear bandits and **real-world datasets** in Section 4.2 and Section 4.3 of the revised submission.
>
> **Q3**: Could you provide more intuition for Lemma 2.14 and clarify the differences in proofs between this one and the previous one?
>
> **A3**: We would like to first provide some intuition behind Lemma 2.14. Let $\\pi\_r \\propto \\exp(\\eta r)$ be the optimal policy under reward function $r$, we consider $J(\\pi\_{r})$ as a function of $r$. Then Taylor expansion gives $J(\\pi\_{g^\*}) - J(\\pi\_{g}) = \\nabla J(\\pi\_{g^\*})(g^\* - g) + \\nabla^2 J(\\pi\_{g\_\\gamma})(g^\* - g)^2$, where $g\_\\gamma = \\gamma g^\* + (1-\\gamma)g$. Since $r^\*$ maximize $J(\\pi\_{r})$, $\\nabla J(\\pi\_{g^\*})=0$. Therefore, $J(\\pi\_{g^\*}) - J(\\pi\_{g})$ can be bounded purely by some second-order term $(g-g^\*)^2$ and a computation gives the form of right-hand-side in Lemma 2.14.
>
> We would like to remark that, our analysis differs from [1] in that how the seemingly arbitrary $\\pi\_{\\gamma}$ is tackled. In [1], the expectation regarding $\\pi\_{\\gamma}$ is directly upper bounded by the expectation regarding $\\pi\_{\\text{ref}}$ multiplied with all-policy concentrability $C^{\\Pi}$. In this work, thanks to pessimism, the expectation regarding $\\pi\_{\\gamma}$ can be controlled by expectation regarding $\\pi^\*$, which means that single-policy-concentrability is sufficient to upper bound the RHS of Lemma 2.15 and therefore get rid of all-policy concentrability.
>
> References:
>
> [1] Zhao, Heyang, et al. "Sharp analysis for kl-regularized contextual bandits and rlhf." arXiv preprint arXiv:2411.04625 (2024).

---

### Official Review · Reviewer_3Ln2 · 2025-11-01

**Soundness:** 3
**Presentation:** 3
**Contribution:** 2
**Rating:** 6
**Confidence:** 3

**Summary:**

The paper studies the sample complexity of regularized policy optimization with general $f$-divergence regularizations in contextual bandits. For the special case of KL-divergence regularization, the authors derive an $\mathcal{O}(\epsilon^{-1})$ upper bound when the reference policy provides sufficient coverage of the optimal regularized policy. They further present a matching lower bound, up to a coverage parameter, that depends on a weaker density-ratio coverage of the optimal regularized policy, indicating that some coverage assumption is necessary for efficient learning. Finally, for $\alpha$-strongly convex function divergences, they establish a minimax-optimal sample complexity that does not depend on any coverage parameters.

**Strengths:**

1. Relaxation from all policy coverage to optimal policy coverage assumption with clever use of pessimistic bonus term
2. The lower bound of Theorem 2.11 has a multiplicative coverage term, showing some coverage assumption is needed for any efficient algorithm
3. The work shows that for $f$-divergence-regularized objectives, if $f$ is strongly convex, then no coverage assumption on the reference policy is necessary.

**Weaknesses:**

1. The worst-case gap between $C^{\pi^\*}$ and $D^2_{\pi^\*}$ scales as $|S||A|$. This linear dependence on the size of the state space can render Algorithm 1 inefficient, even when $C^{\pi^\*}$ is a constant. A more detailed discussion or empirical illustration of how these two coverage measures relate in practice would strengthen the paper.
2. Theorem 3.4 constructs only a specific instance of an $\alpha$-strongly convex $f$ (a scaled $\chi^2$ divergence) that matches the upper bound, rather than establishing a general $f$-dependent lower bound.

**Questions:**

1. The coverage term of lowerbound is bounded by $ \min( C^{\pi^\*}, \mathcal{O}(\exp(\eta) )) $ in Theorem 2.11. Similarly, the coverage term of sample complexity of Algorithm 1 can be bounded by $ \min( D^2_{\pi^\*}, \mathcal{O}(|S||A|\exp(\eta) )) $. Can this be improved to $ \min( D^2_{\pi^\*}, \mathcal{O}(\exp(\eta) )) $?
2. Have the authors tried to show similar upper bound for $f$-divergence-regularized CB when $f$ is only locally strongly-convex by adding an appropriate bonus term similar to Algorithm 1?

---

> ### Author Response · Authors · 2025-11-21
>
> We thank the reviewer for the insightful and constructive feedback. We address the questions as follows.
>
> **Q1**: The worst-case gap between $C^{\\pi^\*}$ and $D^2\_{\\pi^\*}$ scales as $SA$. A more detailed discussion or empirical illustration of how these two coverage measures relate in practice would strengthen the paper.
>
> **A1**: Thank you for your suggestion!
> - In short, $D^2\_{\\pi^\\star}$ is an **instance-dependent** notion, which could also be much smaller than $C^{\\pi^\*}$ in benign cases: See **Proposition B.2** for such a construction.
> - In details, the reward function in practice usually comes from a function class (e.g., neural network), and thus the relation between $D^2\_{\\pi^\*}$ and $C^{\\pi^\*}$ is heavily **instance-dependent**. We take the linear function class as an example. On the one hand, we can construct an instance such that $D^2\_{\\pi^\*} \\approx dC^{\\pi^\*}$ (See our Proposition B.1 for details). On the other hand, we construct an instance where some $(s,a)$ is frequently visited by $\\pi^\*$ but seldomly by $\\pi^{\\text{ref}}$, then this leads to very large $C^{\\pi^\*}$. But we can also ensure that there is some $(s',a')$ such that $g(s',a')$ is very close to $g(s,a)$ for all $g \\in \\mathcal{G}$, and $(s',a')$ is frequetly visited by $\\pi^{\\text{ref}}$. In this case, since the "feature" of $(s,a)$ is frequently visited (by visiting $(s',a')$), $D^2\_{\\pi^\*}$ is small and therefore we have $D^2\_{\\pi^\*} \\ll C^{\\pi^\*}$. These cases show that relation $D^2\_{\\pi^\*}$ and $C^{\\pi^\*}$ varies and can only be determined upon the instance is specified.
> - We have add a more detailed discussion of the illustration above in **Appendix B** of our revised submission.
>
>
>
> **Q2**: Theorem 3.4 constructs only a specific instance of an $\\alpha$-strongly convex $f$.
>
> **A2**: The purpose of Theorem 3.4 is to demonstrate the worst-case hardness of the problem solved in Theorem 3.2. **The existence of one set of hard instances** in the proof of Theorem 3.4 **already certificates the near-optimality of Theorem 3.2**. We think that construct hard instances for general strongly convex $f$ is an interesting direction and we leave it as our future work.
>
> **Q3**: Can sample complexity of Algorithm 1 be improved to $\\min(D^2\_{\\pi^\*}, \\exp(\\eta))$?
>
> **A3**: The sample complexity upper bound of both $D^2\_{\\pi^\*}$ and $\\exp(\\eta)$ are achievable, though through different algorithm. Our analysis already shows that the Algorithm 1 ensures a $D^2\_{\\pi^\*}$-based sample conplexity upper bound. The $\\exp(\\eta)$-dependent sample complexity bound can be obtained by running Algorithm 2 with the $f$-divergence replaced by KL regularization (equivalently, Algorithm 1 without pessimism). Actually, under this algorithm, applying Lemma 2.14 and directly bound $\\mathbb{E}\_{\\pi\_{\\gamma}}$ with $C^{\\Pi}\\mathbb{E}\_{\\pi\_{\\text{ref}}}$, we obtain an $\\eta C^{\\Pi}\\mathbb{E}\_{\\pi\_{\\text{ref}}}[(\\bar{g} - g^\*)^2]$ upper bound of this algorithm. Then, we can bound $C^{\\Pi}$ with $\\exp(\\eta)$ and apply Lemma B.1 and Lemma B.2, yielding the sample complexity upper bound $\\eta\\exp(\\eta)\\log \\mathcal{N}\_{\\epsilon}\\epsilon^{-1}$. We think achieving $\\min(D^2\_{\\pi^\*}, \\exp(\\eta))$ with one single algorithm is a interesting future direction.
>
> **Q4**: Have the authors tried to show similar upper bound for $f$-divergence-regularized CB when $f$ is only locally strongly-convex by adding an appropriate bonus term similar to Algorithm 1?
>
> **A4**: Currently our upper bound analysis for KL-regularized objectives (and consequently for Algorithm 1), especially Lemma 2.14, indeed heavily hinges on the specific closed Gibbs form solution $\\pi|\_r \\propto \\pi\_\\mathrm{ref}(\\cdot)\\exp(\\eta r(\\cdot))$, we leave a more general analysis for locally convex $f$, which is ideally free of the exploitation of the Gibbs form, as an important future direction.

---

### Meta-Review · Area_Chair_iUfM · 2026-01-08

**Summary:**

This paper proves that KL regularization allows offline contextual bandits to achieve a fast $\epsilon^{-1}$ sample complexity while only requiring single-policy coverage. This is an improvement over prior work that suggested such efficiency required covering all possible policies or resulted in a worse $\epsilon^{-2}$ sample complexity.  The work also establish tight sample complexity bound for strongly convex f-divergence.  The problem is relevant (especially for LLM) and the theoretical contribution is clear.  All reviewers are in favor of acceptance.

**Reviewer Concerns:**

Reviewer concerns include:
- Experiments are limited (8QC9, 8Rei, gwK9.) -- addressed:  The authors have added experiments in more general settings.
- The upper and lower bound do not match -- addressed:  There is still technical difficulty, and the paper has sufficiently discussed this issue.

**Reviewer Scores:**

All reviewers give a score of 6.  There is no particular evidence that they will change their scores.

---

### Decision · Program_Chairs · 2026-01-26

Accept (Poster)